# Unsupervised Learning of the Set of Local Maxima

**Lior Wolf**
Facebook AI Research &
The School of Computer Science
Tel Aviv University
wolf@fb.com, wolf@cs.tau.ac.il

**Sagie Benaim & Tomer Galanti**
The School of Computer Science
Tel Aviv University
sagieb@mail.tau.ac.il
tomerga2@post.tau.ac.il

## Abstract

This paper describes a new form of unsupervised learning, whose input is a set of unlabeled points that are assumed to be local maxima of an unknown value function $v$ in an unknown subset of the vector space. Two functions are learned: (i) a set indicator $c$, which is a binary classifier, and (ii) a comparator function $h$ that given two nearby samples, predicts which sample has the higher value of the unknown function $v$. Loss terms are used to ensure that all training samples $x$ are a local maxima of $v$, according to $h$ and satisfy $c(x) = 1$. Therefore, $c$ and $h$ provide training signals to each other: a point $x'$ in the vicinity of $x$ satisfies $c(x) = -1$ or is deemed by $h$ to be lower in value than $x$. We present an algorithm, show an example where it is more efficient to use local maxima as an indicator function than to employ conventional classification, and derive a suitable generalization bound. Our experiments show that the method is able to outperform one-class classification algorithms in the task of anomaly detection and also provide an additional signal that is extracted in a completely unsupervised way.

## 1 Introduction

> ...from so simple a beginning endless forms most beautiful and most wonderful have been, and are being, evolved. (Darwin, 1859)

When we observe the natural world, we see the "most wonderful" forms. We do not observe the even larger quantity of less spectacular forms and we cannot see those forms that are incompatible with existence. In other words, each sample we observe is the result of optimizing some fitness or value function under a set of constraints: the alternative, lower-value, samples are removed and the samples that do not satisfy the constraints are also missing.

The same principle also holds at the sub-cellular level. For example, a gene can have many forms. Some of them are completely synonymous, while others are viable alternatives. The gene forms that become most frequent are those which are not only viable, but which also minimize the energetic cost of their expression (Farkas et al., 2018). For example, the genes that encode proteins comprised of amino acids of higher availability or that require lower expression levels to achieve the same outcome have an advantage. One can expect to observe most often the gene variants that: (i) adhere to a set of unknown constraints ("viable genes"), and (ii) optimize an unknown value function that includes energetic efficiency considerations.

The same idea, of mixing constraints with optimality, also holds for man-made objects. Consider, for example, the set of houses in a given neighborhood. Each architect optimizes the final built form to cope with various aspects, such as the maximal residential floor area, site accessibility, parking considerations, the energy efficiency of the built product, etc. What architects find most challenging, is that this optimization process needs to correspond to a comprehensive set of state and city regulations that regard, for example, the proximity of the built mass of the house to the lot's boundaries, or the compliance of the egress sizes with current fire codes.

In another instance, consider the weights of multiple neural networks trained to minimize the same loss on the same training data, each using a different random initialization. Considering the weights of each trained neural network as a single vector in a sample domain, also fits into the framework of local optimality under constraints. By the nature of the problem, the obtained weights are the local optimum of some loss optimization process. In addition, the weights are sometimes subject to constraints, e.g., by using weight normalization.

The task tackled in this paper is learning the value function and the constraints, by observing only the local maxima of the value function among points that satisfy the constraints. This is an unsupervised problem: no labels are given in addition to the samples.

The closest computational problem in the literature is *one-class classification* (Moya et al., 1993), where one learns a classifier $c$ in order to model a set of unlabeled training samples, all from a single class. In our formulation, two functions are learned: a classifier and a separate value function. Splitting the modeling task between the two, a simpler classifier can be used (we prove this for a specific case) and we also observe improved empirical performance. In addition, we show that the value function, which is trained with different losses and structure from those of the classifier, models a different aspect of the training set. For example, if the samples are images from a certain class, the classifier would capture class membership and the value function would encode image quality. The emergence of a quality model makes sense, since the training images are often homogeneous in their quality.

The classifier $c$ and the value function $v$ provide training signals to each other, in an unsupervised setting, somewhat similar to the way adversarial training is done in GANs (Goodfellow et al., 2014), although the situation between $c$ and $v$ is not adversarial. Instead, both work collaboratively to minimize similar loss functions. Let $\mathbb{S}$ be the set of unlabeled training samples from a space $\mathbb{X}$. Every $\boldsymbol{x} \in \mathbb{S}$ satisfies $c(\boldsymbol{x}) = 1$ for a classifier $c : \mathbb{X} \to \{\pm 1\}$ that models the adherence to the set of constraints (satisfies or not). Alternatively, we can think of $c$ as a class membership function that specifies, if a given input is within the class or not. In addition, we also consider a value function $v$, and for every point $\boldsymbol{x}'$, such that $\|\boldsymbol{x}' - \boldsymbol{x}\| \leq \epsilon$, for a sufficiently small $\epsilon > 0$, we have: $v(\boldsymbol{x}') < v(\boldsymbol{x})$.

This structure leads to a co-training of $v$ and $c$, such that every point $\boldsymbol{x}'$ in the vicinity of $\boldsymbol{x}$ can be used either to apply the constraint $v(\boldsymbol{x}') < v(\boldsymbol{x})$ on $v$, or as a negative training sample for $c$. Which constraint to apply, depends on the other function: if $c(\boldsymbol{x}') = 1$, then the first constraint applies; if $v(\boldsymbol{x}') \geq v(\boldsymbol{x})$, then $\boldsymbol{x}'$ is a negative sample for $c$. Since the only information we have on $v$ pertains to its local maxima, we can only recover it up to an unknown monotonic transformation. We therefore do not learn it directly and instead learn a comparator function $h$, which given two inputs, returns an indication which input has the higher value in $v$.

An alternative view of the learning problem we introduce considers the value function $v$ (or equivalently $h$) as part of a density estimation problem, and not as part of a multi-network game. In this view, $c$ is the characteristic function (of belonging to the support) and $h$ is the comparator of the probability density function (PDF).

## 2 RELATED WORK

The input to our method is a set of unlabeled points. The goal is to model this set. This form of input is shared with the family of methods called one-class classification. The main application of these methods is anomaly detection, i.e., identifying an outlier, given a set of mostly normal (the opposite of abnormal) samples (Chandola et al., 2009).

The literature on one class classification can be roughly divided into three parts. The first includes the classical methods, mostly kernel-base methods, which were applying regularization in order to model the in-class samples in a tight way (Schölkopf et al., 2001). The second group of methods, which follow the advent of neural representation learning, employ classical one-class methods to representations that are learned in an unsupervised way (Hawkins et al., 2002; Sakurada & Yairi, 2014; Xia et al., 2015; Xu et al., 2015; Erfani et al., 2016), e.g., by using autoencoders. Lastly, a few methods have attempted to apply a suitable one-class loss, in order to learn a neural network-based representation from scratch (Ruff et al., 2018). This loss can be generic or specific to a data domain. Recently, Golan & El-Yaniv (2018) achieved state of the art one-class results for visual datasets by training a network to predict the predefined image transformation that is applied to each

of the training images. A score is then used to evaluate the success of this classifier on test images, assuming that out of class images would be affected differently by the image transformations.

Despite having the same structure of the input (an unlabeled training set), our method stands out of the one-class classification and anomaly detection methods we are aware of, by optimizing a specific model that disentangles two aspects of the data: one aspect is captured by a class membership function, similar to many one-class approaches; the other aspect compares pairs of samples. This dual modeling captures the notion that the samples are not nearly random samples from some class, but also the local maximum in this class. While "the local maxima of in-class points" is a class by itself, a classifier-based modeling of this class would require a higher complexity than a model that relies on the structure of the class as pertaining to local maxima, as is proved, for one example, in Sec. A. In addition to the characterization as local maxima, the factorization between the constraints and the values also assists modeling. This is reminiscent of many other cases in machine learning, where a divide and conquer approach reduces complexity. For example, using prior knowledge on the structure of the problem, helps to reduce the complexity in hierarchical models, such as LDA (Blei et al., 2003).

While we use the term "value function", and this function is learned, we do not operate in a reinforcement learning setting, where the term value is often used. Specifically, our problem is not inverse reinforcement learning (Ng & Russell, 2000) and we do not have actions, rewards, or policies.

## 3 Method

Recall that $\mathbb{S}$ is the set of unlabeled training samples, and that we learn two functions $c$ and $v$ such that for all $\boldsymbol{x} \in \mathbb{S}$ it holds that: (i) $c(\boldsymbol{x}) = 1$, and (ii) $\boldsymbol{x}$ is a local maxima of $v$.

For every monotonic function $f$, the setting we define cannot distinguish between $v$, and $f \circ v$. This ambiguity is eliminated, if we replace $v$ by a binary function $h$ that satisfies $h(\boldsymbol{x}, \boldsymbol{x}') = 1$ if $v(\boldsymbol{x}) \geq v(\boldsymbol{x}')$ and $h(\boldsymbol{x}, \boldsymbol{x}') = -1$ otherwise. We found that training $h$ in lieu of $v$ is considerably more stable. Note that we do not enforce transitivity, when training $h$, and, therefore, $h$ can be such that no underlying $v$ exists.

### 3.1 Training $c$ and $h$

When training $c$, the training samples in $\mathbb{S} = \{\boldsymbol{x}_i\}_{i=1}^m$ are positive examples. Without additional constraints, the recovery of $c$ is an ill-posed problem. For example, Ruff et al. (2018) add an additional constraint on the compactness of the representation space. Here, we rely on the ability to generate hard negative points[1]. There are two generators $G_c$ and $G_h$, each dedicated to generating negative training points to either $c$ or $h$, as described in Sec. 3.2 below.

The two generators are conditioned on a positive point $\boldsymbol{x} \in \mathbb{S}$ and each generates one negative point per each $\boldsymbol{x}$: $\boldsymbol{x}' = G_c(\boldsymbol{x})$ and $\boldsymbol{x}'' = G_h(\boldsymbol{x})$. The constraints on the negative points are achieved by multiplying two losses: one pushing $c(\boldsymbol{x}')$ to be negative, and the other pushing $h(\boldsymbol{x}'', \boldsymbol{x})$ to be negative.

Let $\ell(p, y) := -\frac{1}{2}((y+1) \log(p) + (1-y) \log(1-p))$ be the binary cross entropy loss for $y \in \{\pm 1\}$. $c$ and $h$ are implemented as neural networks trained to minimize the following losses, respectively:

$$\mathcal{L}_C := \frac{1}{m} \sum_{\boldsymbol{x} \in \mathbb{S}} \ell(c(\boldsymbol{x}), 1) + \frac{1}{m} \sum_{\boldsymbol{x} \in \mathbb{S}} \ell(c(G_c(\boldsymbol{x})), -1) \cdot \ell(h(G_c(\boldsymbol{x}), \boldsymbol{x}), -1) \tag{1}$$

$$\mathcal{L}_H := \frac{1}{m} \sum_{\boldsymbol{x} \in \mathbb{S}} \ell(h(\boldsymbol{x}, \boldsymbol{x}), 1) + \frac{1}{m} \sum_{\boldsymbol{x} \in \mathbb{S}} \ell(c(G_h(\boldsymbol{x})), -1) \cdot \ell(h(G_h(\boldsymbol{x}), \boldsymbol{x}), -1) \tag{2}$$

The first sum in $\mathcal{L}_C$ ensures that $c$ classifies all positive points as positive. The second sum links the outcome of $h$ and $c$ for points generated by $G_c$. It is given as a multiplication of two losses. This multiplication encourages $c$ to focus on the cases where $h$ predicts with a higher probability that the point $G_c(\boldsymbol{x})$ is more valued than $\boldsymbol{x}$.

---

[1]"hard negative" is a terminology often used in the object detection and boosting literature, which means negative points that challenge the training process.

---

**Algorithm 1** Training $c$ and $h$

---

**Require:** $\mathbb{S}$: positive training points; $\lambda$: a trade-off parameter; $T$: number of epochs.
 1: Initialize $c, h, G_c$ and $G_h$ randomly.
 2: **for** $i = 1, ..., T$ **do**
 3:     Train $G_c$ for one epoch to minimize $-\mathcal{L}_C$
 4:     Train $c$ for one epoch to minimize $\mathcal{L}_C$
 5:     Train $G_h$ for one epoch to minimize $\frac{\lambda}{m} \sum_{\boldsymbol{x} \in \mathbb{S}} ||\boldsymbol{x} - G_h(\boldsymbol{x})|| - \mathcal{L}_H$
 6:     Train $h$ for one epoch to minimize $\mathcal{L}_H$
 7: **return** $c, h$

---

The first term of $\mathcal{L}_C$ (respectively $\mathcal{L}_H$) depends on $c$'s (respectively $h$'s) parameters only. The second term of $\mathcal{L}_C$ (respectively $\mathcal{L}_H$), however, depends on both $h$'s and $c$'s parameters as well as $G_c$'s (respectively $G_h$'s) parameters.

The loss $\mathcal{L}_H$ is mostly similar. It ensures that $h$ has positive values when the two inputs are the same, at least at the training points. In addition, it ensures that for the generated negative points $\boldsymbol{x}'$, $h(\boldsymbol{x}', \boldsymbol{x})$ is $-1$, especially when $c(\boldsymbol{x}')$ is high.

One can alternatively use a symmetric $\mathcal{L}_H$, by including an additional term $\frac{1}{m} \sum_{\boldsymbol{x} \in \mathbb{S}} \ell(c(G_h(\boldsymbol{x})), -1) \cdot \ell(h(\boldsymbol{x}, G_h(\boldsymbol{x})), 1)$. This, in our experiments, leads to very similar results, and we opt for the slightly simpler version.

## 3.2 Negative Point Generation

We train two generators, $G_c$ and $G_h$, to produce hard negative samples for the training of $c$ and $h$, respectively. The two generators both receive a point $\boldsymbol{x} \in \mathbb{S}$ as input, and generate another point in the same space $\mathbb{X}$. They are constructed using an encoder-decoder architecture, see Sec. 3.4 for the exact specifications.

When training $G_c$, the loss $-\mathcal{L}_C$ is minimized. In other words, $G_c$ finds, in an adversarial way, points $x'$, that maximize the error of $c$ (the first term of $\mathcal{L}_C$ does not involve $G_c$ and does not contribute, when training $G_c$).

$G_h$ minimizes during training the loss $\frac{\lambda}{m} \sum_{\boldsymbol{x} \in \mathbb{S}} ||\boldsymbol{x} - G_h(\boldsymbol{x})|| - \mathcal{L}_H$, for some parameter $\lambda$. Here, in addition to the adversarial term, we add a term that encourages $G_h(\boldsymbol{x})$ to be in the vicinity of $\boldsymbol{x}$. This is added, since the purpose of $h$ is to compare nearby points, allowing for the recovery of points that are local maxima. In all our experiments we set $\lambda = 1$.

The need for two generators, instead of just one, is verified in our ablation analysis, presented in Sec. 4. One may wonder why two are needed. One reason stems from the difference in the training loss: $h$ is learned locally, while $c$ can be applied anywhere. In addition, $c$ and $h$ are challenged by different points, depending on their current state during training. By the structure of the generators, they only produce one point per input $\boldsymbol{x}$, which is not enough to challenge both $c$ and $h$.

## 3.3 Training Procedure

The training procedure follows the simple interleaving scheme presented in Alg. 1. We train the networks in turns: $G_c$ and then $c$, followed by $G_h$ and then $h$. Since the datasets in our experiments are relatively small, each turn is done using all mini-batches of the training dataset $\mathbb{S}$. The ADAM optimization scheme is used with mini-batches of size 32.

The training procedure has self regularization properties. For example, assuming that $G_h(\boldsymbol{x}) \neq \boldsymbol{x}$, $\mathcal{L}_H$ as a function of $h$, has a trivial global minima. This solution is to assign $h(\boldsymbol{x}', \boldsymbol{x})$ to 1 iff $\boldsymbol{x}' = \boldsymbol{x}$. However, for this specific $h$, the only way for $G_h$ to maximize $L_H$ is to rely on $c$ and $h$ being smooth and to select points $\boldsymbol{x}' = G_h(\boldsymbol{x})$ that converge to $\boldsymbol{x}$, at least for some points in $\boldsymbol{x} \in \mathbb{S}$. In this case, both $\ell(c(G_h(\boldsymbol{x})), -1)$ and $\ell(h(G_h(\boldsymbol{x}), \boldsymbol{x}), -1)$ will become high, since $c(\boldsymbol{x}') \approx 1$ and $h(\boldsymbol{x}', \boldsymbol{x}) \approx 1$.

### 3.4 Architecture

In the image experiments (MNIST, CIFAR10 and GTSRB), the neural networks $G_h$ and $G_c$ employ the DCGAN architecture of Radford et al. (2015). This architecture consists of an encoder-decoder type structure, where both the encoder and the decoder have five blocks. Each encoder (resp. decoder) block consists of a 2-strided convolution (resp. deconvolution) followed by a batch norm layer, and a ReLU activation. The fifth decoder block consists of a 2-strided convolution followed by a tanh activation instead. $c$ and $h$'s architectures consist of four blocks of the same structure as for the encoder. This is followed by a sigmoid activation.

For the Cancer Genome Atlas experiment, each encoder (resp. decoder) block consists of a fully connected (FC) layer, a batch norm layer and a Leaky Relay activation (slope of 0.2). Two blocks are used for the encoder and decoder. The encoder's first FC layer reduces the dimension to 512 and the second to 256. The decoder is built to mirror this. $c$ and $h$ consist of two blocks, where the first FC layer reduces the dimension to 512 and the second to 1. This is followed by a sigmoid activation.

### 3.5 Analysis

In Appendix A, We show an example in which modeling using local-maxima-points is an efficient way to model, in comparison to the conventional classification-based approach. We then extend the framework of spectral-norm bounds, which were derived in the context of classification, to the case of unsupervised learning using local maxima.

## 4 Experiments

Since we share the same form of input with one-class classification, we conduct experiments using one-class classification benchmarks. These experiments both help to understand the power of our model in capturing a given set of samples, as well as study the properties of the two underlying functions $c$ and $h$.

Following acceptable benchmarks in the field, specifically the experiments done by Ruff et al. (2018), we consider single classes out of multiclass benchmarks, as the basis of one-class problems. For example, in MNIST, the set $\mathbb{S}$ is taken to be the set of all training images of a particular digit. When applying our method, we train $h$ and $c$ on this set. To clarify: there are no negative samples during training.

Post training, we evaluate both $c$ and $h$ on the one class classification task: positive points are now the MNIST test images of the same digit used for training, and negative points are the test images of all other digits. This is repeated ten times, for digits 0–9. In order to evaluate $h$, which is a binary function, we provide it with two replicas of the test point.

The classification ability is evaluated as the AUC obtained on this classification task. The same experiment was conducted for CIFAR-10 where instead of digits we consider the ten different class labels. The results are reported in Tab. 1, which also states the literature baseline values reported by Ruff et al. (2018). As can be seen, for both CIFAR-10 and MNIST, $c$ strongly captures class-membership, outperforming the baseline results in most cases. $h$ is less correlated with class membership, resulting in much lower mean AUC values and higher standard deviations. However, it should not come as a surprise that $h$ does contain such information.

Indeed, the difference in shape (single input vs. two inputs) between $c$ and $h$ makes them different but not independent. $c$, as a classifier, strongly captures class membership. We can expect $h$, which compares two samples, to capture relative properties. In addition, $h$, due to the way negative samples are collected, is expected to model local changes, at a finer resolution than $c$. Since it is natural to expect that the samples in the training set would provide images that locally maximize some clarity score, among all local perturbations, one can expect quality to be captured by $h$.

To test this hypothesis, we considered positive points to be test points of the relevant one-class, and negative points to be points with varying degree of Gaussian noise added to them. We then measure using AUC, the ability to distinguish between these two classes.

Table 1: One class experiments on the MNIST and CIFAR-10 datasets. For MNIST, there is one experiment per digit, where the training samples are the training set of this digit. The reported numbers are the AUC for classifying one-vs-rest, using the test set of this digit vs. the test sets of all other digits. For CIFAR-10, the same experiment is run with a class label, instead of the digits. Reported numbers (in all tables) are averaged over 10 runs with random initializations. Each reported value is the mean result $\pm$ the standard deviation.

| Digit | KDE (Parzen, 1962) | AnoGAN (Schlegl, 2017) | Deep SVDD (Ruff et al., 2018) | Our $c$ | Our $h$ |
|---|---|---|---|---|---|
| 0 | 97.1±0.0 | 96.6±1.3 | 98.0±0.7 | **99.1**±0.2 | 83.5±11.6 |
| 1 | 98.9±0.0 | 99.2±0.6 | **99.7**±0.1 | 97.2±0.7 | 50.7±25.7 |
| 2 | 79.0±0.0 | 85.0±2.9 | 91.7±0.8 | **91.9**±0.4 | 67.1±15.7 |
| 3 | 86.2±0.0 | 88.7±2.1 | 91.9±1.5 | **94.3**±0.7 | 62.4±25.9 |
| 4 | 87.9±0.0 | 89.4±1.3 | **94.9**±0.8 | 94.2±0.3 | 85.7±10.7 |
| 5 | 73.8±0.0 | 88.3±2.9 | **88.5**±0.9 | 87.2±2.0 | 73.3±14.5 |
| 6 | 87.6±0.0 | 94.7±2.7 | 98.3±0.5 | **98.8**±0.2 | 62.8±15.9 |
| 7 | 91.4±0.0 | 93.5±1.8 | **94.6**±0.9 | 93.9±0.5 | 61.6±10.8 |
| 8 | 79.2±0.0 | 84.9±2.1 | 93.9±1.6 | **96.0**±0.1 | 45.8±17.7 |
| 9 | 88.2±0.0 | 92.4±1.1 | 96.5±0.3 | **96.7**±0.3 | 66.8±14.5 |
| Airplane | 61.2±0.0 | 67.1±2.5 | 61.7±4.2 | **74.0**±1.2 | 48.9±13.1 |
| Automobile | 64.0±0.0 | 54.1±3.4 | 65.9±2.1 | **74.7**±1.6 | 64.6±5.4 |
| Bird | 50.1±0.0 | 52.9±3.0 | 50.8±0.8 | **62.8**±2.0 | 53.2±4.5 |
| Cat | 56.4±0.0 | 54.5±1.9 | **59.1**±1.4 | 57.2±2.0 | 51.4±6.6 |
| Deer | 66.2±0.0 | 65.1±3.2 | 60.9±1.1 | **67.8**±2.2 | 55.0±9.3 |
| Dog | 62.4±0.0 | 60.3±2.6 | **65.7**±2.5 | 60.2±1.6 | 58.9±3.7 |
| Frog | 74.9±0.0 | 58.5±1.4 | 67.7±2.6 | **75.3**±3.9 | 60.7±4.5 |
| Horse | 62.6±0.0 | 62.5±0.8 | 67.3±0.9 | **68.5**±2.8 | 58.1±3.8 |
| Ship | 75.1±0.0 | 75.8±4.1 | 75.9±1.2 | **78.1**±1.2 | 66.9±7.1 |
| Truck | 76.0±0.0 | 66.5±2.8 | 73.1±1.2 | **79.5**±1.5 | 70.3±8.3 |

Table 2: An ablation analysis on the ten CIFAR classes (shown in order, Airplane to Truck).

| | 1 | 2 | 3 | 4 | 5 | 6 | 7 | 8 | 9 | 10 |
|---|---|---|---|---|---|---|---|---|---|---|
| Baseline $c$ | **74.0** | **74.7** | **62.8** | 57.2 | **67.8** | 60.2 | **75.3** | 68.5 | **78.1** | **79.5** |
| Baseline $h$ | 48.9 | 64.6 | 53.2 | 51.4 | 55.0 | 58.9 | 60.7 | 58.1 | 66.9 | 70.3 |
| $c$ only | 73.0 | 63.8 | 59.1 | **59.6** | 60.4 | 60.7 | 62.8 | 62.1 | 77.2 | 73.3 |
| $h$ only | 35.6 | 51.9 | 50.1 | 48.0 | 48.3 | 48.0 | 68.0 | 54.7 | 75.6 | 73.1 |
| $c$ with $G_c$ only | 73.4 | 74.3 | 61.2 | 58.8 | 66.4 | 59.0 | 72.7 | 70.3 | 77.1 | 75.1 |
| $h$ with $G_c$ only | 63.7 | 68.3 | 59.2 | 56.6 | 58.8 | 57.4 | 60.7 | 65.5 | 71.3 | 74.2 |
| $c$ with $G_h$ only | 73.2 | 71.2 | 59.6 | 51.7 | 65.4 | 60.9 | 68.3 | **68.9** | 76.7 | 77.2 |
| $h$ with $G_h$ only | 56.0 | 65.3 | 55.5 | 53.2 | 50.6 | 58.6 | 54.8 | 58.4 | 65.2 | 71.8 |

As can be seen in Fig. 1, $h$ is much better at identifying noisy images than $c$, for all noise levels. This property is class independent, and in Fig. 3 (Appendix C), we repeat the experiment for all test images (not just from the one class used during training), observing the same phenomenon.

We employ CIFAR also to perform an ablation analysis comparing the baseline method's $c$ and $h$ with four alternatives: (i) training $c$ without training $h$, employing only $G_c$; (ii) training $h$ and $G_h$ without training $c$ nor $G_c$; (iii) training both $h$ and $c$ but using only the $G_c$ generator to obtain negative samples to both networks; and (iv) training both $h$ and $c$ but using only the $G_h$ generator for both. The results, which can be seen in Tab. 2, indicate that the complete method is superior to the variants, since it outperform these in the vast majority of the experiments.

Next, we evaluate our method on data from the German Traffic Sign Recognition (GTSRB) Benchmark of Houben et al. (2013). The dataset contains 43 classes, from which one class (stop signs, class #15) was used by Ruff et al. (2018) to demonstrate one-class classification where the negative class is the class of adversarial samples (presumably based on a classifier trained on all 43 classes).

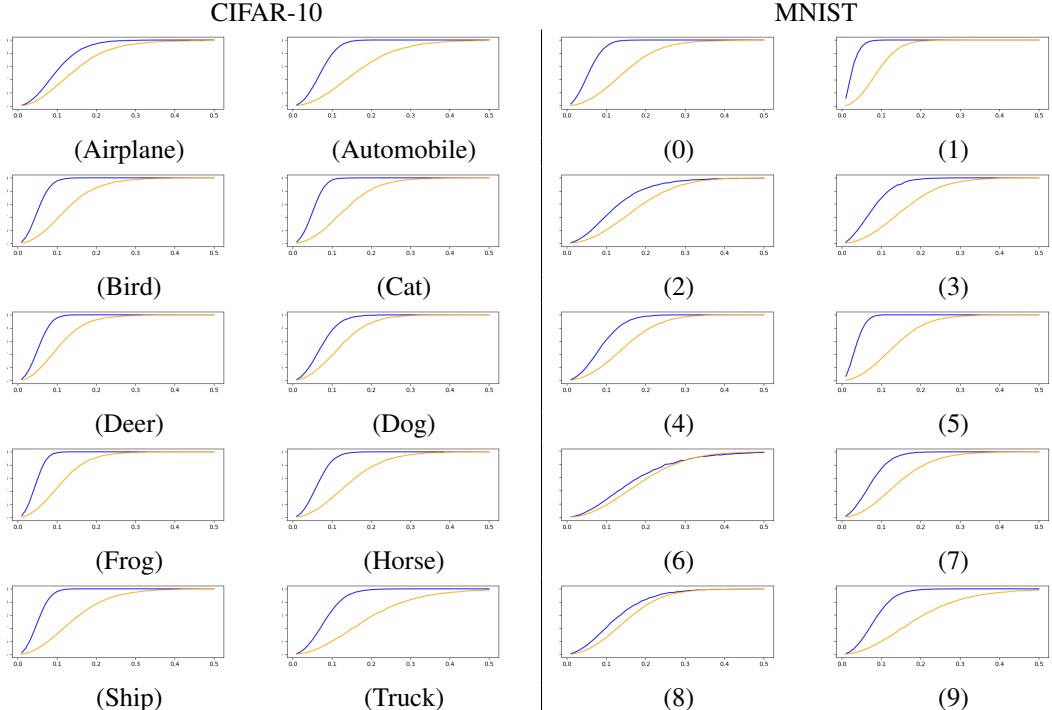

Figure 1: The ability to differentiate between an in-class image and an in-class image with added noise for both $c$ (yellow) and $h$ (blue). The x-axis is the amount of noise (SD of the Gaussian noise). The y-axis is the AUC. As can be seen, for both CIFAR-10 and MNIST, $h$ is much more attuned to the image quality.

We were not able to obtain these samples by the time of the submission. Instead, We employ the sign data in order to evaluate three other one-class tasks: (i) the conventional task, in which a class is compared to images out of all other 42 classes; (ii) class image vs. noise image, as above, using Gaussian noise with a fixed noise level of $\sigma = 0.2$; (iii) same as (ii) only that after training on one class, we evaluate on images from all classes.

The results are presented, for the first 20 classes of GTSRB, in Tab. 3. The reported results are an average over 10 random runs. On the conventional one-class task (i), both our $c$ and $h$ neural networks outperform the baseline Deep-SVDD method, with $c$ performing better than $h$, as in the MNIST and CIFAR experiments. Also following the same pattern as before, the results indicate that $h$ captures image noise better than both $c$ and Deep-SVDD, for both the test images of the training class and the test images from all 43 classes.

In order to explore the possibility of using the method out of the context of one-class experiments and for scientific data analysis, we downloaded samples from the Cancer Genome Atlas (`https://cancergenome.nih.gov/`). The data contains mRNA expression levels for over 22,000 genes, measured from the blood of 9,492 cancer patients. For most of the patients, there is also survival data in days. We split the data to 90% train and 10% test.

We run our method on the entire train data and try to measure whether the functions recovered are correlated with the survival data on the test data. While, as mentioned in Sec. 1, the gene expression optimizes a fitness function, and one can claim that gene expressions that are less fit, indicate an expected shortening in longevity, this argument is speculative. Nevertheless, since survival is the only regression signal we have, we focus on this experiment.

We compare five methods: (i) the $h$ we recover, (ii) the $c$ we recover, (iii) the $h$ we recover, when learning only $h$ and not $c$, (iv) the $c$ we recover, when learning only $c$ and not $h$, (v) the first PCA of the expression data, (vi) the classifier of DeepSVDD. The latter is used as baseline due to the shared form of input with our method. However, we do not perform an anomaly detection experiment.

Table 3: Results obtained on the GTSRB dataset on three one-class tasks. Reported are AUC values in percents. DS denotes Deep-SVDD by Ruff et al. (2018).

| Class | (i) Multiclass | | | (ii) Noise in-class | | | (iii) Noise all images. | | |
|---|---|---|---|---|---|---|---|---|---|
| | $c$ | $h$ | DS | $c$ | $h$ | DS | $c$ | $h$ | DS |
| 1 | 92.6 | 77.8 | 86.2 | 61.1 | 62.3 | 61.8 | 55.1 | 58.9 | 44.7 |
| 2 | 78.0 | 75.4 | 71.9 | 75.6 | 96.3 | 74.7 | 71.4 | 92.3 | 51.4 |
| 3 | 78.3 | 79.5 | 65.8 | 71.0 | 95.0 | 66.1 | 79.0 | 98.5 | 50.0 |
| 4 | 79.7 | 81.7 | 63.9 | 89.1 | 97.0 | 66.3 | 71.0 | 82.0 | 53.2 |
| 5 | 79.7 | 79.3 | 73.2 | 90.1 | 95.6 | 48.7 | 72.3 | 84.5 | 56.3 |
| 6 | 73.8 | 66.4 | 81.8 | 91.1 | 85.3 | 88.1 | 75.3 | 75.2 | 62.0 |
| 7 | 91.0 | 90.2 | 73.6 | 93.0 | 94.1 | 84.1 | 58.1 | 72.4 | 55.2 |
| 8 | 82.1 | 75.4 | 74.6 | 93.7 | 93.9 | 51.6 | 71.0 | 82.1 | 56.7 |
| 9 | 80.2 | 84.7 | 73.4 | 92.4 | 93.7 | 54.3 | 70.5 | 81.0 | 53.8 |
| 10 | 85.8 | 74.9 | 79.2 | 82.0 | 93.4 | 88.7 | 71.0 | 84.0 | 57.7 |
| 11 | 81.9 | 81.7 | 82.7 | 93.4 | 93.9 | 65.0 | 78.2 | 78.4 | 68.3 |
| 12 | 86.9 | 84.6 | 54.3 | 78.3 | 92.6 | 89.8 | 70.3 | 89.1 | 64.5 |
| 13 | 88.1 | 82.1 | 60.0 | 84.0 | 91.2 | 74.6 | 78.2 | 79.1 | 60.5 |
| 14 | 93.5 | 93.7 | 57.6 | 82.3 | 85.4 | 78.9 | 76.0 | 77.4 | 63.4 |
| 15 | 98.2 | 93.7 | 71.9 | 67.3 | 81.2 | 65.0 | 54.0 | 64.0 | 49.2 |
| 16 | 87.6 | 90.5 | 71.8 | 59.0 | 78.3 | 90.0 | 55.3 | 63.2 | 55.6 |
| 17 | 92.5 | 96.8 | 76.7 | 73.1 | 83.4 | 83.1 | 58.3 | 67.2 | 55.6 |
| 18 | 99.3 | 85.4 | 64.4 | 73.0 | 92.1 | 77.7 | 87.3 | 97.2 | 50.7 |
| 19 | 79.5 | 79.7 | 52.2 | 68.1 | 81.2 | 90.4 | 62.0 | 78.3 | 57.8 |
| 20 | 92.9 | 92.9 | 52.1 | 76.3 | 78.2 | 81.6 | 52.3 | 63.0 | 74.0 |
| Avg | **86.1** | 83.3 | 69.4 | 79.7 | **88.2** | 74.0 | 68.3 | **78.4** | 57.0 |

Table 4: Correlation between the recovered functions and the patient's survival.

| | Local | | Standard | |
|---|---|---|---|---|
| Method | Pearson correlation | P-value | Pearson correlation | P-value |
| Our $h$ | 0.076 | **0.021** | 0.041 | 0.384 |
| Our $c$ | 0.020 | 0.520 | 0.029 | 0.444 |
| Our $h$ trained without $c$ | 0.033 | 0.405 | 0.017 | 0.716 |
| Our $c$ trained without $h$ | 0.029 | 0.444 | 0.031 | 0.420 |
| First PCA of mRNA expression | 0.047 | 0.308 | 0.006 | 0.903 |
| Deep-SVDD | 0.021 | 0.510 | 0.032 | 0.410 |

In the simplest experiment, we treat $h$ as a unary function by replicating the single input, as done above. We call this the standard correlation experiment. However, $h$ was trained in order to compare two local points and we, therefore, design the local correlation protocol. First, we identify for each test datapoint, the closest test point. We then measure the difference in the target value (the patient's survival) between the two datapoints, the difference in value for unary functions (e.g., for $c$ or for the first PCA), or $h$ computed for the two datapoints. This way vectors of the length of the number of test data points are obtained. We use the Pearson correlation between these vectors and the associated p-values as the test statistic.

The results are reported in Tab. 4. As can be seen, the standard correlation is low for all methods. However, for local correlation, which is what $h$ is trained to recover, the $h$ obtained when learning both $h$ and $c$ is considerably more correlated than the other options, obtaining a significant p-value of 0.021. Interestingly, the ability to carve out parts of the space with $c$, when learning $h$ seems significant and learning $h$ without $c$ results in a much reduced correlation.

Finally, we test our method on the Gaussian Mixture Model data following Balduzzi et al. (2018), who perform a similar experiment in order to study the phenomenon of mode hopping. In this experiment, the data is sampled from 16 Gaussians placed on a 4x4 grid with coordinates $-1.5, 0.5,$

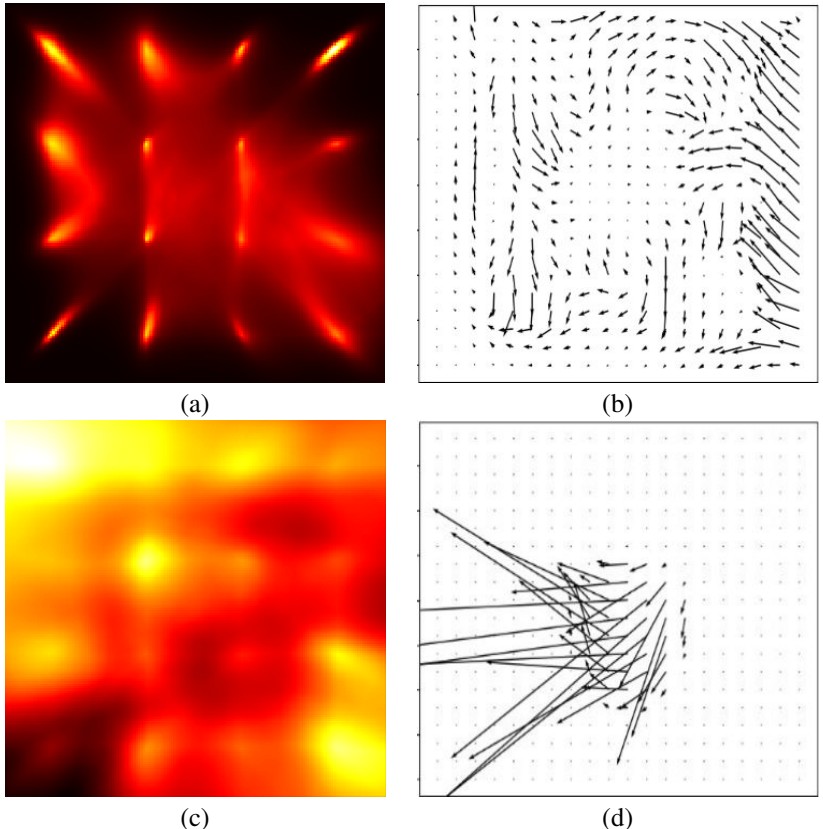

Figure 2: Results for a mixture of Gaussians placed on a 2D grid, following Balduzzi et al. (2018). (a) The values of the function $c$ across the 2D domain, when $c$ and $h$ are jointly trained. (b) The comparator $h$ shown as a quiver plot of the direction of maximal increase in value. (c) The values of $c$ when it is trained alone without $h$. (d) A quiver plot for $h$, when it is trained without $c$.

0.5 and 1.5 in each axis. In our case, since we model local maxima, we take each Gaussian to have a standard deviation that is ten times smaller than that of Balduzzi et al. (2018): 0.01 instead of 0.1. We treat the mixture as a single class and sample a training set from it, to which we apply our methods as well as the variants where each network trains separately.

The results are depicted in Fig. 2, where we present both $c$ and $h$. As can be seen, our complete method captures with $c$ the entire distribution, while training $c$ without $h$ runs leads to an unstable selection of a subset of the modes. Similarly, training $h$ without $c$ leads to an $h$ function that is much less informative than the one extracted when the two networks are trained together.

## 5 DISCUSSION

The current machine learning literature focuses on models that are smooth almost everywhere. The label of a sample is implicitly assumed as likely to be the same as those of the nearby samples. In contrast to this curve-based world view, we focus on the cusps. This novel world view could be beneficial also in supervised learning, e.g., in the modeling of sparse events.

Our model recovers two functions: $c$ and $h$, which are different in form. This difference may be further utilized to allow them to play different roles post learning. Consider, e.g., the problem of drug design, in which one is given a library of drugs. The constraint function $c$ can be used, post training, to filter a large collection of molecules, eliminating toxic or unstable ones. The value function $h$ can be used as a local optimization score in order to search locally for a better molecule.

ACKNOWLEDGEMENTS

This project has received funding from the European Research Council (ERC) under the European Unions Horizon 2020 research and innovation programme (grant ERC CoG 725974). The contribution of Sagie Benaim is part of Ph.D. thesis research conducted at Tel Aviv University.

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

# A   ANALYSIS

We show an example in which modeling using local-maxima-points is an efficient way to model, in comparison to the conventional classification-based approach. We then extend the framework of spectral-norm bounds, which were derived in the context of classification, to the case of unsupervised learning using local maxima.

## A.1   MODELING USING $\arg\max v$ IS BENEFICIAL

While modeling with a classifier $c$ is commonplace, modeling a set $\mathbb{S} = \{x_i\}_{i=1}^m$ as the local maxima of a function is much less conventional. Next, we will argue that at least in some situations, it may be advantageous.

We compare the complexity of a ReLU neural network $\boldsymbol{W}_2\phi(\boldsymbol{W}_1 x + \boldsymbol{b})$ modeling a set $\mathbb{S} = \{x_i\}_{i=1}^m$ of $m$ real numbers as either a classifier or as maxima of a value function. Here, $\boldsymbol{W}_1$ and $\boldsymbol{W}_2$ are linear transformations, $\boldsymbol{b}$ is a vector and $\phi(u_1, \ldots, u_n) := (\max(0, u_1), \ldots, \max(0, u_n))$ is the ReLU activation function, for $u_1, \ldots, u_n \in \mathbb{R}$ and $n \in \mathbb{N}$. We denote by $c_{\mathbb{S}} : \mathbb{R} \to \{\pm 1\}$ the function that satisfies, $c_{\mathbb{S}}(x) = 1$ if and only if $x \in \mathbb{S}$.

For this purpose, we take any distribution $D$ over $[x_1 - 1, x_m + 1]$ that has positive probability for sampling from $\mathbb{S}$. Formally, $D$ is a mixture distribution that samples at probability $q > 0$ from $D_0$ and probability $1 - q$ from $D_1$, where $D_0$ is a distribution supported by $\mathbb{S}$ and $D_1$ is a distribution supported by the segment $[x_1 - 1, x_m + 1]$. The task is to achieves error $\leq \epsilon$ in approximating $c_{\mathbb{S}}$. The error of a function $c : \mathbb{R} \to \mathbb{R}$ is measured by $\mathbb{1}_{x \sim D}[c(x) \neq c_{\mathbb{S}}(x)]$, which is the probability of $c$ incorrectly labeling a random number $x \sim D$.

We show that there is a ReLU neural network $v(x) = \boldsymbol{W}_2\phi(\boldsymbol{W}_1 x + \boldsymbol{b})$ with $2m$ neurons, such that, the set of local maxima of $v$ is $\mathbb{S}$. In particular, we have: $\mathbb{E}_{x \sim D}\mathbb{1}[c_v(x) \neq c_{\mathbb{S}}(x)] = 0$. Here, $c_v : \mathbb{R} \to \{\pm 1\}$, that satisfies, $c_v(x) = 1$ if and only if $x$ is a local maxima of $v$. Additionally,

we show that any such $v$ has at least $2m - 1$ hidden neurons. On the other hand, we show that for any distribution $D$ (with the specifications above), any classification neural network $c(x) = W_2\phi(W_1x + b)$ that has error $\mathbb{E}_{x\sim D}\mathbb{1}[c(x) \neq c_\mathbb{S}(x)] \leq \epsilon$ has at least $3m$ hidden neurons.

**Theorem 1.** *Let $\mathbb{S} = \{x_i\}_{i=1}^m \subset \mathbb{R}$ be any set of points such that $x_i < x_{i+1}$ for all $i \in \{1, \ldots, m - 1\}$. We define $c_\mathbb{S} : \mathbb{R} \to \{\pm 1\}$ to be the function, such that $c_\mathbb{S}(x) = 1$ if and only if $x \in \mathbb{S}$. Then,*

1. *There is a ReLU neural network $v : \mathbb{R} \to \mathbb{R}$ of the form $v(x) = W_2\phi(W_1x + b)$ with $2m$ hidden neurons such that the set of local maximum points of $v$ is $\mathbb{S}$.*

2. *Any ReLU neural network $v : \mathbb{R} \to \mathbb{R}$ of the form $v(x) = W_2\phi(W_1x + b)$ such that any $x \in \mathbb{S}$ is a local maxima of $v$ has at least $2m - 1$ neurons.*

3. *Let $D = q \cdot D_0 \cup (1 - q) \cdot D_1$ be a distribution that samples at probability $q > 0$ from $D_0$ and probability $1 - q$ from $D_1$, where $D_0$ is a distribution supported by $\mathbb{S}$ and $D_1$ is a distribution supported by the segment $[x_1 - 1, x_m + 1]$. Then, for a small enough $\epsilon > 0$, every ReLU neural network $c : \mathbb{R} \to \mathbb{R}$ of the form $c(x) = W_2\phi(W_1x + b)$, such that $\mathbb{E}_{x\sim D}\mathbb{1}[c(x) \neq c_\mathbb{S}(x)] \leq \epsilon$ has at least $3m$ hidden neurons.*

*Proof.* We begin by proving (1). We construct $v$ as follows:

- $\forall x \in (-\infty, x_1]: v(x) = x - x_1 + 1$.

- $\forall i \in \{1, \ldots, m - 1\} : \forall x \in [x_i, \frac{x_i + x_{i+1}}{2}]: v(x) = \frac{-2}{x_{i+1} - x_i}(x - x_i) + 1$.

- $\forall i \in \{1, \ldots, m - 1\} : \forall x \in [\frac{x_i + x_{i+1}}{2}, x_{i+1}]: v(x) = \frac{1}{x_{i+1} - x_i}(x - x_{i+1}) + 1$.

- $\forall x \in [x_m, \infty): v(x) = x_m - x + 1$.

we consider that $v$ is a piece-wise linear function with $2m$ linear pieces and $\arg\max v = \mathbb{S}$. By Thm. 2.2 in Arora et al. (2018), this function can be represented as a ReLU neural network of the form $v(x) = W_2\phi(W_1x + b)$, that has $2m$ hidden neurons.

Next, we prove (2). Let $v : \mathbb{R} \to \mathbb{R}$ be a function of the form $v(x) = W_2\phi(W_1x + b)$, such that each $x \in \mathbb{S}$ is a local maxima of it. First, by Thm. 2.1 in Arora et al. (2018), $v$ is a piece-wise linear function. We claim that $v$ has at least two linear pieces between each consecutive points $x_i$ and $x_{i+1}$, for $i \in \{1, \ldots, m - 1\}$. Assume the contrary, i.e., there is an index $i \in \{1, \ldots, m - 1\}$, such that, $v$ has only one piece between $x_i$ and $x_{i+1}$. If $v(x_i) = v(x_{i+1})$, then, $v$ is constant between $x_i$ and $x_{i+1}$, and therefore, $x_i$ and $x_{i+1}$ are not local maximas of $v$, in contradiction. If $v(x_i) < v(x_{i+1})$, then, because $v$ is linear between $x_i$ and $x_{i+1}$, for every point $x \in (x_i, x_{i+1})$, we have $v(x_i) < v(x)$, in contradiction to the assumption that $x_i$ is a local maxima of $v$. If $v(x_i) > v(x_{i+1})$, then, because $v$ is linear between $x_i$ and $x_{i+1}$, for every point $x \in (x_i, x_{i+1})$, we have $v(x_{i+1}) < v(x)$, in contradiction to the assumption that $x_{i+1}$ is a local maxima of $v$. We conclude that $v$ has at least two linear pieces between the points $x_i$ and $x_{i+1}$, for all $i \in \{1, \ldots, m - 1\}$. Therefore, $v$ has at least $2m$ pieces. By Thm. 2.2 in Arora et al. (2018), $v$ has at least $2m - 1$ hidden neurons.

Next, we prove (3). We denote $x_0 = x_1 - 1$ and $x_{m+1} = x_m + 1$. Let $\mathbb{P}_{D_0}[x_i]$ be the probability of sampling $x_i$ from $D_0$. Since $D_0$ is supported by $\mathbb{S}$, we have: $q \cdot \mathbb{P}_{D_0}[x_i] > 0$. We define $\alpha := q\min_{i\in\{1,\ldots,m\}} \mathbb{P}_{D_0}[x_i]$. In addition, $D_1$ is a continuous distribution supported by the closed segment $[x_1 - 1, x_m + 1]$. Thus, by Weierstrass' extreme value theorem, the probability density function $\mathbb{P}_{D_1}[x]$ of $D_1$ that is a continuous function, obtains its extreme values within the segment. In addition, since $D_1$ is supported by $[x_1 - 1, x_m + 1]$, we have: $\mathbb{P}_{D_1}[x] > 0$ for all $x \in [x_1 - 1, x_m + 1]$. By combining the above two statements, we conclude that there is a point $x^* \in [x_1 - 1, x_m + 1]$ such that $\mathbb{P}_{D_1}[x] \geq \mathbb{P}_{D_1}[x^*] > 0$ for every $x \in [x_1 - 1, x_m + 1]$. We denote by $\beta := (1 - q)\min_{i\in\{0,\ldots,m\}} \mathbb{P}_{D_1}[x \in (x_i, x_{i+1})] > 0$. Since we are interested in proving the claim for a small enough $\epsilon > 0$, we can simply assume that $\epsilon < \min(\alpha, \beta)$ and $c : \mathbb{R} \to \mathbb{R}$ a ReLU neural network of the form $c(x) = W_2\phi(W_1x + b)$.

We have:

$$\mathbb{E}_{x \sim D} \mathbb{1}\left[c(x) \neq c_{\mathbb{S}}(x)\right] = q \mathbb{E}_{x \sim D_0} \mathbb{1}\left[c(x) \neq c_{\mathbb{S}}(x)\right] + (1-q) \mathbb{E}_{x \sim D_1} \mathbb{1}\left[c(x) \neq c_{\mathbb{S}}(x)\right]$$

$$\geq q \mathbb{E}_{x \sim D_0} \mathbb{1}\left[c(x) \neq c_{\mathbb{S}}(x)\right] \geq \alpha \sum_{i=1}^{m} \mathbb{1}\left[c(x) \neq c_{\mathbb{S}}(x)\right] \tag{3}$$

Assume by contradiction that: $c(x_i) \neq c_{\mathbb{S}}(x_i)$. Then, $\mathbb{E}_{x \sim D} \mathbb{1}\left[c(x) \neq c_{\mathbb{S}}(x)\right] \geq \alpha > \epsilon$ in contradiction. Therefore, $c(x_i) = c_{\mathbb{S}}(x_i) = 1$ for every $x_i \in \mathbb{S}$.

We also have:

$$\mathbb{E}_{x \sim D} \mathbb{1}\left[c(x) \neq c_{\mathbb{S}}(x)\right] = q \mathbb{E}_{x \sim D_0} \mathbb{1}\left[c(x) \neq c_{\mathbb{S}}(x)\right] + (1-q) \mathbb{E}_{x \sim D_1} \mathbb{1}\left[c(x) \neq c_{\mathbb{S}}(x)\right]$$

$$\geq (1-q) \mathbb{E}_{x \sim D_1} \mathbb{1}\left[c(x) \neq c_{\mathbb{S}}(x)\right] \tag{4}$$

Assume by contradiction that there is $i \in \{1, \ldots, m-1\}$, such that the set $E_i = \{x \in (x_i, x_{i+1}) | c(x) = 0\}$ is finite. Then,

$$\mathbb{E}_{x \sim D} \mathbb{1}\left[c(x) \neq c_{\mathbb{S}}(x)\right] \geq (1-q) \mathbb{E}_{x \sim D_1} \mathbb{1}\left[c(x) \neq c_{\mathbb{S}}(x)\right]$$

$$\geq (1-q) \mathbb{P}_{D_1}\left[x \in (x_i, x_{i+1})\right] \geq \beta > \epsilon \tag{5}$$

in contradiction. Let $i \in \{1, \ldots, m-1\}$, $a_i$ and $b_i$ be two points such that $x_i < a_i < b_i < x_{i+1}$ and $c(a_i) = c(b_i) = 0$. Since $c$ is a continuous function and piece-wise linear and the four points $(x_i, 1), (a, 0), (b, 0), (x_{i+1}, 1)$ are not co-linear, we conclude that $c$ has at least three linear pieces in the segment $[x_i, x_{i+1}]$. Similarly, $c$ has at least two linear pieces in each of the segments $[x_1 - 1, x_1]$ and $[x_m, x_m + 1]$. We conclude that $c$ has at least $3m + 1$ pieces. By Thm. 2.2 in Arora et al. (2018), $c$ has at least $3m$ hidden neurons. □

In the above theorem we showed that there exists a ReLU neural network $v(x) = \boldsymbol{W}_2 \phi(\boldsymbol{W}_1 x + \boldsymbol{b})$ with $2m$ hidden neurons that captures the set $\mathbb{S}$ as its local maximas. Furthermore, we note that the set of functions that satisfy these conditions (i.e., shallow ReLU neural networks with $2m$ hidden neurons that capture the set $\mathbb{S}$) is relatively limited. For instance, any such $v$ behaves as a piece-wise linear function between any $x_i$ and $x_{i+1}$ with only two linear pieces. Therefore, any such $v$ is uniquely determined by the set $\mathbb{S}$ up to some freedom in the selection of the linear pieces between $x_i$ and $x_{i+1}$ in $\mathbb{S}$. On the other hand, a shallow ReLU neural network $v(x) = \boldsymbol{W}_2 \phi(\boldsymbol{W}_1 x + \boldsymbol{b})$ with more than $2m$ hidden neurons is capable of having more than two linear pieces between any $x_i$ and $x_{i+1}$.

## A.2 A GENERALIZATION BOUND

The following lemma provides a generalization bound that expresses the generalization of learning $c$ along with $v$. In the following generalization bound, we assume there is an arbitrary distribution $D$ of positive samples. In addition, we parameterize the class, $\mathcal{V} = \{v_\theta : \mathbb{R}^d \to \mathbb{R} \mid \theta \in \Theta\}$, of value functions by vectors of parameters $\theta \in \Theta$ and the class, $\mathcal{C} = \{\text{sign} \circ f_\omega : \mathbb{R}^d \to \{\pm 1\} \mid \omega \in \Omega\}$, of classifiers by $\omega \in \Omega$. We upper bound the probability of a mistake done by any classifier $c_\omega \in \mathcal{C}$ and value function $v_\theta \in \mathcal{V}$ on a random sample $\boldsymbol{x} \sim D$. In this case, $v_\theta$ and $c_\omega$ mistake if $\boldsymbol{x}$ is not a local maxima of $v_\theta$ or classified as negative by $c_\omega$. The upper bound is decomposed into the sum of the average error of $c_\omega$ and $v_\theta$ on a dataset $\mathbb{S} \overset{\text{i.i.d}}{\sim} D^m$ and a regularization term.

See Appendix B for the exact formulation and the proof.

**Lemma 1** (Informal). *Let $\mathcal{V} = \{v_\theta : \mathbb{R}^d \to \mathbb{R} \mid \theta \in \Theta\}$ be a class of value functions and $\mathcal{C} = \{\text{sign} \circ f_\omega : \mathbb{R}^d \to \{\pm 1\} \mid \omega \in \Omega\}$ a class of classifiers. Assume that $v_\theta$ and $f_\omega$ are ReLU neural networks of fixed architectures, with parameters $\theta$ and $\omega$ (resp.). Let $C(g)$ is the spectral complexity of the neural network $g$ and $N_\epsilon(\boldsymbol{x}) := \{\boldsymbol{u} \in \mathbb{R}^d \mid \|\boldsymbol{u} - \boldsymbol{x}\|_2 \leq \epsilon\}$ an $\epsilon$-neighborhood of $\boldsymbol{x}$. Let $D$ be a distribution of positive examples. With probability at least $1 - \delta$ over the selection of*

*the data* $\mathbb{S} = \{x_i\}_{i=1}^m \overset{\text{i.i.d}}{\sim} D^m$, *for every* $v_\theta \in \mathcal{V}$ *and* $c_\omega \in \mathcal{C}$, *we have:*

$$\mathbb{P}_{\boldsymbol{x}} \left[ v_\theta(\boldsymbol{x}) \neq \max_{\boldsymbol{u} \in N_\epsilon(\boldsymbol{x})} v_\theta(\boldsymbol{u}) \text{ or } c_\omega(\boldsymbol{x}) \neq 1 \right]$$

$$\leq \frac{1}{m} \sum_{i=1}^m \boldsymbol{1} \left[ v_\theta(\boldsymbol{x}_i) \neq \max_{\boldsymbol{u} \in N_\epsilon(\boldsymbol{x}_i)} v_\theta(\boldsymbol{u}) \text{ or } c_\omega(\boldsymbol{x}_i) \neq 1 \right] \tag{6}$$

$$+ \mathcal{O} \left( \sqrt{\frac{C(v_\theta) + C(f_\omega) + \log\left(\frac{m}{\delta}\right)}{m}} \right)$$

The above lemma shows that the probability of $\boldsymbol{x} \sim D$ to be a local maxima of $v_\theta$ and classified as a positive example by $c_\omega$, is at most the sum of the probability of $\boldsymbol{x} \in \mathbb{S}$ to be a local maxima of $v_\theta$ and classified as a positive example by $c_\omega$ and a penalty term. The penalty in this case is of the form $\mathcal{O} \left( \sqrt{\frac{C(v_\theta)+C(f_\omega)+\log(m/\delta)}{m}} \right)$, where $m$ is the number of examples in the dataset and $C(v_\theta) + C(f_\omega)$ is the sum of the spectral norms of $v_\theta$ and $f_\omega$. This suggests a tradeoff between the sum of the spectral complexities of $v_\theta$ and $f_\omega$ and the ability to generalize. The bound is similar asymptotically to the bounds of Neyshabur et al. (2018) and Bartlett et al. for (multi-class) supervised classification. In their bound, the penalty term is of the form $\mathcal{O} \left( \sqrt{\frac{C(f)+\log\left(\frac{m}{\delta}\right)}{m}} \right)$, where the (multi-class) classifier is of the form $c(\boldsymbol{x}) = \arg\max_{i \in \{1,\ldots,t\}} f(\boldsymbol{x})_i$, for a neural network $f : \mathbb{R}^d \to \mathbb{R}^t$.

Our analysis focused on the value $v_\theta$ and not on the comparator $h$. However, the complexities of the two are expected to be similar, since a value function can be converted to a comparator by employing $h(\boldsymbol{x}_1, \boldsymbol{x}_2) = \text{sign}(v_\theta(\boldsymbol{x}_1) - v_\theta(\boldsymbol{x}_2))$.

## B A FORMAL STATEMENT OF THE GENERALIZATION BOUND

In this section, we build upon the theory presented by Neyshabur et al. (2018) and provide a generalization bound that expresses the guarantees of learning $c$, along with $v$ for a specific setting.

Before we introduce the generalization bound, we introduce the necessary terminology and setup. We assume that the sampling space is a ball of radius $B$, i.e., $\mathbb{X} = \mathbb{X}_{B,d} := \{\boldsymbol{x} \in \mathbb{R}^d \mid ||\boldsymbol{x}||_2 \leq B\}$. Each value function $v_\theta \in \mathcal{V}$ is a ReLU neural network of the form $v_\theta(\boldsymbol{x}) = \boldsymbol{W}_r \phi(\boldsymbol{W}_{r-1}\phi(\ldots\phi(\boldsymbol{W}_1\boldsymbol{x}))$, where, $\boldsymbol{W}_i \in \mathbb{R}^{d_i \times d_{i+1}}$ for $i \in \{1,\ldots,r\}$ such that $d_{r+1} := 1$ and $d_1 := d$. In addition, $\phi(\boldsymbol{x}) = (\max(0, x_1), \ldots, \max(0, x_n))$ is the ReLU activation function extended to all $n \in \mathbb{N}$ and $\boldsymbol{x} \in \mathbb{R}^n$. We denote, $\theta = (\boldsymbol{W}_1, \ldots, \boldsymbol{W}_r)$. The set $\mathcal{C}$ consists of classifiers $c_\omega := \text{sign} \circ f_\omega$ such that each function $f_\omega : \mathbb{R}^d \to \mathbb{R}$ is a ReLU neural network of the form $f_\omega(\boldsymbol{x}) = \boldsymbol{U}_s \phi(\boldsymbol{U}_{s-1}\phi(\ldots\phi(\boldsymbol{U}_1\boldsymbol{x}))$, where, $\boldsymbol{U}_i \in \mathbb{R}^{d'_i \times d'_{i+1}}$ for $i \in \{1,\ldots,s\}$ such that $e_{s+1} := 1$ and $d'_1 := d$. We denote $\omega = (\boldsymbol{U}_1, \ldots, \boldsymbol{U}_s)$. Additionally, we denote, $q_1 := \max\{d_i\}_{i=1}^{r+1}$ and $q_2 := \max\{d'_i\}_{i=1}^{s+1}$.

The spectral complexity of a ReLU neural network $g_\beta = \boldsymbol{V}_k \phi(\boldsymbol{V}_{k-1}\phi(\ldots\phi(\boldsymbol{V}_1\boldsymbol{x}))$ with parameters $\beta = (\boldsymbol{V}_1, \ldots, \boldsymbol{V}_k)$ is defined as follows:

$$C(g_\beta) := C(\beta) := \prod_{i=1}^k ||\boldsymbol{W}_i||_2^2 \sum_{i=1}^k \frac{||\boldsymbol{W}_i||_F^2}{||\boldsymbol{W}_i||_2^2} \tag{7}$$

For two distributions $P$ and $Q$ over a set $\mathbb{X}$, we denote the KL-divergence between them by, $D_{\text{KL}}(Q||P) := \mathbb{E}_{\boldsymbol{x} \sim Q}[\log(Q(\boldsymbol{x})/P(\boldsymbol{x}))]$. For two functions function $f, g : \mathbb{R} \to \mathbb{R}$, we denote the asymptotic symbols: $g(x) = \mathcal{O}(f(x))$ to specify that $g(x) \leq c \cdot f(x)$, for some constant $c > 0$. We denote by $\boldsymbol{1}[x]$ the indicator, if a boolean $x \in \{\text{true}, \text{false}\}$ is true or false.

We define a margin loss $\ell_{\gamma_1, \gamma_2} : \mathbb{X} \times \Omega \times \Theta \to \mathbb{R}$ of the form:

$$\ell_{\gamma_1, \gamma_2}(\boldsymbol{x}; \omega, \theta) := \boldsymbol{1} \left[ v_\theta(\boldsymbol{x}) < \max_{\boldsymbol{u} \in N_\epsilon(\boldsymbol{x})} v_\theta(\boldsymbol{u}) - \gamma_1 \text{ or } \text{sign}(f_\omega(\boldsymbol{x}) - \gamma_2) \neq 1 \right] \tag{8}$$

where, $\gamma_1, \gamma_2 > 0$ are fixed margins and $N_\epsilon(\boldsymbol{x}) := \{\boldsymbol{u} \mid ||\boldsymbol{u} - \boldsymbol{x}||_2 \leq \epsilon\}$ is the $\epsilon$-neighborhood of $\boldsymbol{x}$, for a fixed $\epsilon > 0$. In this model, the margins serve as parameters that dictate the amount of tolerance in classifying an example as positive. Similar to the standard learning framework, for a fixed distribution $D$ over $\mathbb{X}$, the goal of a learning procedure is to return (given some input) $\omega$ and $\theta$ that minimize the following generalization risk function:

$$F_D[\omega, \theta] := \mathbb{E}_{\boldsymbol{x} \sim D}[\ell_{0,0}(\boldsymbol{x}; \omega, \theta)] \tag{9}$$

The learning process has no direct access to the distribution $D$. Instead, it is provided with a set of $m$ i.i.d samples from $D$, $\mathbb{S} = \{\boldsymbol{x}_i\}_{i=1}^m \overset{\text{i.i.d}}{\sim} D^m$. In order to estimate the generalization risk, the empirical risk function is used during training:

$$\hat{F}_{\mathbb{S}}^{\gamma_1, \gamma_2}[\omega, \theta] := \frac{1}{m} \sum_{i=1}^m \ell_{\gamma_1, \gamma_2}(\boldsymbol{x}_i; \omega, \theta) \tag{10}$$

The following lemma provides a generalization bound that expresses the generalization of learning $c$ along with $h$.

**Lemma 2.** *Let $\mathbb{X} := \mathbb{X}_{B,n}$, $\mathcal{V}$ and $\mathcal{C}$ be as above. Let $D$ be a distribution of positive examples. With probability at least $1 - \delta$ over the selection of the data $\mathbb{S} = \{x_i\}_{i=1}^m \overset{\text{i.i.d}}{\sim} D^m$, for every $v_\theta \in \mathcal{V}$ and $c_\omega \in \mathcal{C}$, we have:*

$$F_D[\theta, \omega] \leq \hat{F}_{\mathbb{S}}^{\gamma_1, \gamma_2}[\theta, \omega] + \mathcal{O}\left(\sqrt{\frac{B^2\left[r^2 q_1 \log(rq_1)\frac{C(v_\theta)}{\gamma_1^2} + s^2 q_2 \log(sq_2)\frac{C(f_\omega)}{\gamma_2^2}\right] + \log\left(\frac{m}{\delta}\right)}{m}}\right) \tag{11}$$

### B.1 PROOF OF LEM. 2

All over the proofs, we will make use of two generic classes of functions $\mathcal{G} = \{g_\theta : \mathbb{X} \to \mathbb{R}^2 \mid \theta \in \Theta\}$ and $\mathcal{H} = \{h_\omega : \mathbb{X} \to \mathbb{R}^2 \mid \omega \in \Omega\}$. For simplicity, we denote the indices of $g_\theta(\boldsymbol{x})$ and $h_\omega(\boldsymbol{x})$ by $-1$ and $1$ (instead of $1$ and $2$). Given a target function $y : \mathbb{X} \to \{\pm 1\}$ and two functions $g_\theta : \mathbb{X} \to \mathbb{R}^2$ and $h_\omega : \mathbb{X} \to \mathbb{R}^2$, we denote the loss of them with respect to a sample $\boldsymbol{x}$ by:

$$
\begin{aligned}
e_{\gamma_1, \gamma_2}(\boldsymbol{x}; \theta, \omega) := & \mathbb{1}\left[g_\theta(\boldsymbol{x})[-y(\boldsymbol{x}_i)] - \gamma_1 > g_\theta(\boldsymbol{x})[y(\boldsymbol{x}_i)]\right] \\
& \vee \mathbb{1}\left[h_\omega(\boldsymbol{x})[-y(\boldsymbol{x}_i)] - \gamma_2 > h_\omega(\boldsymbol{x})[y(\boldsymbol{x}_i)]\right]
\end{aligned} \tag{12}
$$

The generalization risk:

$$L^{\gamma_1, \gamma_2}[\theta, \omega] := \mathbb{E}_{\boldsymbol{x} \sim D}[e_{\gamma_1, \gamma_2}(\boldsymbol{x}; \theta, \omega)] \tag{13}$$

And the empirical risk:

$$\hat{L}^{\gamma_1, \gamma_2}[\theta, \omega] := \frac{1}{m} \sum_{i=1}^m e_{\gamma_1, \gamma_2}(\boldsymbol{x}_i; \theta, \omega) \tag{14}$$

We modify the proof of Lem. 1 in Neyshabur et al. (2018), such that it will fit our purposes.

**Lemma 3.** *Let $y : \mathbb{X} \to \{\pm 1\}$ be a target function. Let $\mathcal{G} = \{g_\theta : \mathbb{X} \to \mathbb{R}^2 \mid \theta \in \Theta\}$ and $\mathcal{H} = \{h_\omega : \mathbb{X} \to \mathbb{R}^2 \mid \omega \in \Omega\}$ be two classes class of functions (not necessarily neural networks). Let $P_1$ and $P_2$ be any two distributions on the parameters $\Theta$ and $\Omega$ (resp.) that are independent of the training data. Then, for any $\gamma_1, \gamma_2, \delta > 0$, with probability $\geq 1 - \delta$ over the training set of size $m$, for any two posterior distributions $q_\theta$ and $q_\omega$ over $\Theta$ and $\Omega$ (resp.), such that $\mathbb{P}_{\theta', \omega'}[|g_{\theta'}(\boldsymbol{x}) - g_\theta(\boldsymbol{x})|_\infty \leq \frac{\gamma_1}{4}$ and $|h_{\omega'}(\boldsymbol{x}) - h_\omega(\boldsymbol{x})|_\infty \leq \frac{\gamma_2}{4}] \geq \frac{1}{2}$, we have:*

$$L_{0,0}[\theta, \omega] \leq \hat{L}_{\gamma_1, \gamma_2}[\theta, \omega] + 4\sqrt{\frac{D_{\text{KL}}(q_\theta || P_1) + D_{\text{KL}}(q_\omega || P_2) + \log(\frac{6m}{\delta})}{m - 1}} \tag{15}$$

*Proof.* Let $S_{\theta, \omega}^{\gamma_1, \gamma_2} \subset \Theta \times \Omega$ be a set with the following properties:

$$S_{\theta, \omega}^{\gamma_1, \gamma_2} = \left\{(\theta', \omega') \in \Theta \times \Omega \mid \forall \boldsymbol{x} \in \mathbb{X} : |g_{\theta'}(\boldsymbol{x}) - g_\theta(\boldsymbol{x})|_\infty < \frac{\gamma_1}{4} \text{ and } |h_{\omega'}(\boldsymbol{x}) - h_\omega(\boldsymbol{x})|_\infty < \frac{\gamma_2}{4}\right\} \tag{16}$$

We construct a distribution $\tilde{Q}$ over $\Theta \times \Omega$, with probability density function:

$$\tilde{q}(\theta', \omega') = \frac{1}{Z} \begin{cases} q_\theta(\theta') \cdot q_\omega(\omega') & \text{if } (\theta', \omega') \in S_{\theta,\omega}^{\gamma_1,\gamma_2} \\ 0 & \text{otherwise} \end{cases} \tag{17}$$

Here, $Z$ is a normalizing constant. By the assumption in the lemma, $Z = \mathbb{P}[(\theta', \omega') \in S_{\theta,\omega}^{\gamma_1,\gamma_2}] \geq \frac{1}{2}$. By the definition of $\tilde{Q}$, we have:

$$\max_{\boldsymbol{x} \in \mathbb{X}} |g_{\theta'}(\boldsymbol{x}) - g_\theta(\boldsymbol{x})|_\infty < \frac{\gamma_1}{4} \text{ and } \max_{\boldsymbol{x} \in \mathbb{X}} |h_{\omega'}(\boldsymbol{x}) - h_\omega(\boldsymbol{x})|_\infty < \frac{\gamma_2}{4} \tag{18}$$

Therefore,

$$\max_{\boldsymbol{x} \in \mathbb{X}} \left| |g_{\theta'}(\boldsymbol{x})[-1] - g_{\theta'}(\boldsymbol{x})[1]| - |g_\theta(\boldsymbol{x})[-1] - g_\theta(\boldsymbol{x})[1]| \right| < \frac{\gamma_1}{2} \tag{19}$$

and also,

$$\max_{\boldsymbol{x} \in \mathbb{X}} \left| |h_{\omega'}(\boldsymbol{x})[-1] - h_{\omega'}(\boldsymbol{x})[1]| - |h_\omega(\boldsymbol{x})[-1] - h_\omega(\boldsymbol{x})[1]| \right| < \frac{\gamma_2}{2} \tag{20}$$

Since this equation holds uniformly for all $\boldsymbol{x} \in \mathbb{X}$, we have:

$$\begin{aligned} L_{0,0}[\theta, \omega] &\leq L_{\frac{\gamma_1}{2}, \frac{\gamma_2}{2}}[\theta', \omega'] \\ \hat{L}_{\frac{\gamma_1}{2}, \frac{\gamma_2}{2}}[\theta', \omega'] &\leq \hat{L}_{\gamma_1, \gamma_2}[\theta, \omega] \end{aligned} \tag{21}$$

Now using the above inequalities together with Eq. 6 in Mcallester (2003), with probability $1 - \delta$ over the training set we have:

$$\begin{aligned} L_{0,0}(\theta, \omega) &\leq \mathbb{E}_{\theta', \omega'} L_{\frac{\gamma_1}{2}, \frac{\gamma_2}{2}}[\theta', \omega'] \\ &\leq \mathbb{E}_{\theta', \omega'} \hat{L}_{\frac{\gamma_1}{2}, \frac{\gamma_2}{2}}[\theta', \omega'] + 2\sqrt{\frac{2(D_{\mathrm{KL}}(\tilde{q}||P_1 \times P_2) + \log(\frac{2m}{\delta}))}{m - 1}} \\ &\leq \hat{L}_{\gamma_1, \gamma_2}[\theta, \omega] + 2\sqrt{\frac{2(D_{\mathrm{KL}}(\tilde{q}||P_1 \times P_2) + \log(\frac{2m}{\delta}))}{m - 1}} \\ &\leq \hat{L}_{\gamma_1, \gamma_2}[\theta, \omega] + 4\sqrt{\frac{D_{\mathrm{KL}}(q_\theta \times q_\omega||P_1 \times P_2) + \log(\frac{6m}{\delta})}{m - 1}} \end{aligned} \tag{22}$$

where the last inequality follows from the following observation.

Let $S^c$ denote the complement set of $S_{\theta,\omega}^{\gamma_1,\gamma_2}$ and $\tilde{q}^c$ denote the density function $q := q_\theta \times q_\omega$ restricted to $S^c$ and normalized. In addition, we denote $p := P_1 \times P_2$. Then,

$$D_{\mathrm{KL}}(q||p) = Z D_{\mathrm{KL}}(\tilde{q}||p) + (1 - Z) D_{\mathrm{KL}}(\tilde{q}^c||p) - H(Z) \tag{23}$$

where $H(Z) = -Z \log Z - (1 - Z) \log(1 - Z) \leq 1$ is the binary entropy function. Since the KL-divergence is always positive, we get,

$$D_{\mathrm{KL}}(\tilde{q}||p) = \frac{1}{Z}[D_{\mathrm{KL}}(q||p) + H(Z) - (1 - Z)D_{\mathrm{KL}}(\tilde{q}^c||p)] \leq 2(D_{\mathrm{KL}}(q||p) + 1) \tag{24}$$

Since $P_1 \times P_2$ are $q_\theta \times q_\omega$ are independent joint distributions, we have: $D_{\mathrm{KL}}(q_\theta \times q_\omega||P_1 \times P_2) = D_{\mathrm{KL}}(q_\theta||P_1) + D_{\mathrm{KL}}(q_\omega||P_2)$. $\qquad \square$

**Lemma 4.** *Let $\mathcal{V} = \{v_\theta : \mathbb{X} \to [0, 1] \mid \theta \in \Theta\}$ be a class of value functions $v_\theta(\boldsymbol{x}) \in [0, 1]$ and $\mathcal{C} = \{c_\omega = \mathrm{sign} \circ f_\omega \mid f_\omega : \mathbb{X} \to \mathbb{R}, \omega \in \Omega\}$ a class of classifiers (not necessarily neural networks). We define two classes of functions $\mathcal{G} = \{g_\theta = (\max_{\boldsymbol{u} \in N_\epsilon(\boldsymbol{x})} v_\theta(\boldsymbol{u}), v_\theta(\boldsymbol{x})) \mid \theta \in \Theta\}$ and $\mathcal{H} = \{h_\omega = (0, f_\omega(\boldsymbol{x})) \mid \omega \in \Omega\}$. Then,*

$$\begin{aligned} \mathbb{P}_{\theta', \omega'} &\left[ \max_{\boldsymbol{x} \in \mathbb{X}} |g_{\theta'}(\boldsymbol{x}) - g_\theta(\boldsymbol{x})|_\infty < \frac{\gamma_1}{4} \text{ and } \max_{\boldsymbol{x} \in \mathbb{X}} |h_{\omega'}(\boldsymbol{x}) - h_\omega(\boldsymbol{x})|_\infty < \frac{\gamma_2}{4} \right] \\ &\geq \mathbb{P}_{\theta'} \left[ \max_{\boldsymbol{x} \in \mathbb{X}} |v_{\theta'}(\boldsymbol{x}) - v_\theta(\boldsymbol{x})| < \frac{\gamma_1}{4} \right] \cdot \mathbb{P}_{\omega'} \left[ \max_{\boldsymbol{x} \in \mathbb{X}} |f_{\omega'}(\boldsymbol{x}) - f_\omega(\boldsymbol{x})| < \frac{\gamma_2}{4} \right] \end{aligned} \tag{25}$$

*where, $\theta' \sim q_\theta$ and $\omega' \sim q_\omega$.*

*Proof.* We would like to prove that $\max_{\boldsymbol{x} \in \mathbb{X}} |g_{\theta'}(\boldsymbol{x}) - g_\theta(\boldsymbol{x})|_\infty \leq \frac{\gamma_1}{4}$ if $\max_{\boldsymbol{x} \in \mathbb{X}} |v_{\theta'}(\boldsymbol{x}) - v_\theta(\boldsymbol{x})| \leq \frac{\gamma_1}{4}$ and $|h_{\omega'}(\boldsymbol{x}) - h_\omega(\boldsymbol{x})|_\infty \leq \frac{\gamma_2}{4}$ if $|f_{\omega'}(\boldsymbol{x}) - f_\omega(\boldsymbol{x})| \leq \frac{\gamma_2}{4}$. Since $\theta'$ and $\omega'$ are independent, it will prove the desired inequality.

First, we consider that:

$$
\begin{aligned}
|g_{\theta'}(\boldsymbol{x}) - g_\theta(\boldsymbol{x})|_\infty &= \max \left( \left| \max_{\boldsymbol{u} \in N_\epsilon(\boldsymbol{x})} v_{\theta'}(\boldsymbol{x}) - \max_{\boldsymbol{u} \in N_\epsilon(\boldsymbol{x})} v_\theta(\boldsymbol{x}) \right|, |v_{\theta'}(\boldsymbol{x}) - v_\theta(\boldsymbol{x})| \right) \\
&\leq \max \left( \left| \max_{\boldsymbol{u} \in N_\epsilon(\boldsymbol{x})} v_{\theta'}(\boldsymbol{x}) - \max_{\boldsymbol{u} \in N_\epsilon(\boldsymbol{x})} v_\theta(\boldsymbol{x}) \right|, \frac{\gamma_1}{4} \right)
\end{aligned}
\tag{26}
$$

With no loss of generality, we assume that $\max_{\boldsymbol{u} \in N_\epsilon(\boldsymbol{x})} v_{\theta'}(\boldsymbol{x}) \geq \max_{\boldsymbol{u} \in N_\epsilon(\boldsymbol{x})} v_\theta(\boldsymbol{x})$ and denote $\boldsymbol{x}^* = \arg \max_{\boldsymbol{u} \in N_\epsilon(\boldsymbol{x})} v_{\theta'}(\boldsymbol{x})$. Therefore, we have:

$$
\begin{aligned}
\left| \max_{\boldsymbol{u} \in N_\epsilon(\boldsymbol{x})} v_{\theta'}(\boldsymbol{x}) - \max_{\boldsymbol{u} \in N_\epsilon(\boldsymbol{x})} v_\theta(\boldsymbol{x}) \right| &= \max_{\boldsymbol{u} \in N_\epsilon(\boldsymbol{x})} v_{\theta'}(\boldsymbol{x}) - \max_{\boldsymbol{u} \in N_\epsilon(\boldsymbol{x})} v_\theta(\boldsymbol{x}) \\
&= v_{\theta'}(\boldsymbol{x}^*) - \max_{\boldsymbol{u} \in N_\epsilon(\boldsymbol{x})} v_\theta(\boldsymbol{x}) \\
&\leq v_{\theta'}(\boldsymbol{x}^*) - v_\theta(\boldsymbol{x}^*) \leq \frac{\gamma_1}{4}
\end{aligned}
\tag{27}
$$

Next, we consider that:

$$
|h_{\omega'}(\boldsymbol{x}) - h_\omega(\boldsymbol{x})|_\infty = \max \left( |0 - 0|, |f_{\omega'}(\boldsymbol{x}) - f_\omega(\boldsymbol{x})| \right) \leq \frac{\gamma_2}{4}
\tag{28}
$$

$\square$

**Lemma 5.** *Let $\mathcal{V} = \{v_\theta : \mathbb{X} \to [0,1] \mid \theta \in \Theta\}$ be a class of value functions $v_\theta(\boldsymbol{x}) \in [0,1]$ and $\mathcal{C} = \{c_\omega = \text{sign} \circ f_\omega \mid f_\omega : \mathbb{X} \to \mathbb{R}, \omega \in \Omega\}$ a class of classifiers (not necessarily neural networks). Let $P_1$ and $P_2$ be any two distributions over the parameters $\Theta$ and $\Omega$ (resp.) that are independent of the training data. Then, for any $\gamma_1, \gamma_2, \delta > 0$, with probability $\geq 1 - \delta$ over the training set of size $m$, for any two posterior distributions $q_\theta$ and $q_\omega$ over $\Theta$ and $\Omega$ (resp.), such that $\mathbb{P}_{\theta' \sim q_\theta}[|v_{\theta'}(\boldsymbol{x}) - v_\theta(\boldsymbol{x})| \leq \frac{\gamma_1}{4}] \geq \frac{1}{\sqrt{2}}$ and $\mathbb{P}_{\omega' \sim q_\omega}[|f_{\omega'}(\boldsymbol{x}) - f_\omega(\boldsymbol{x})| \leq \frac{\gamma_2}{4}] \geq \frac{1}{\sqrt{2}}$, we have:*

$$
\begin{aligned}
&\mathbb{E}_{\boldsymbol{x} \sim D} \boldsymbol{1} \left[ \max_{\boldsymbol{u} \in N_\epsilon(\boldsymbol{x})} v_\theta(\boldsymbol{u}) > v_\theta(\boldsymbol{x}) \text{ or } \text{sign}(f_\omega(\boldsymbol{x})) \neq 1 \right] \\
\leq &\frac{1}{m} \sum_{i=1}^m \boldsymbol{1} \left[ \max_{\boldsymbol{u} \in N_\epsilon(\boldsymbol{x})} v_\theta(\boldsymbol{u}) - \gamma_1 > v_\theta(\boldsymbol{x}_i) \text{ or } \text{sign}(f_\omega(\boldsymbol{x}_i) - \gamma_2) \neq 1 \right] \\
&+ 4 \sqrt{\frac{D_{\text{KL}}(q_\theta \| P_1) + D_{\text{KL}}(q_\omega \| P_2) + \log(\frac{6m}{\delta})}{m - 1}}
\end{aligned}
\tag{29}
$$

*Proof.* Let $\mathcal{G}$ and $\mathcal{H}$ be as in Lem. 4. By Lem. 4 and our assumption,

$$
\mathbb{P}_{\theta', \omega'} \left[ \max_{\boldsymbol{x} \in \mathbb{X}} |g_{\theta'}(\boldsymbol{x}) - g_\theta(\boldsymbol{x})|_\infty < \frac{\gamma_1}{4} \text{ and } \max_{\boldsymbol{x} \in \mathbb{X}} |h_{\omega'}(\boldsymbol{x}) - h_\omega(\boldsymbol{x})|_\infty < \frac{\gamma_2}{4} \right] \geq \frac{1}{2}
\tag{30}
$$

We note that all of the samples in $D$ are positive. Therefore, by Lem. 3, with probability at least $1 - \delta$, we have:

$$
\begin{aligned}
&\mathbb{E}_{\boldsymbol{x} \sim D} \boldsymbol{1} \left[ g_\theta(\boldsymbol{x})[-1] > g_\theta(\boldsymbol{x})[1] \text{ or } h_\omega(\boldsymbol{x})[-1] > h_\omega(\boldsymbol{x})[1] \right] \\
\leq &\frac{1}{m} \sum_{i=1}^m \boldsymbol{1} \left[ g_\theta(\boldsymbol{x}_i)[-1] - \gamma_1 > g_\theta(\boldsymbol{x}_i)[1] \text{ or } h_\omega(\boldsymbol{x}_i)[-1] - \gamma_2 > h_\omega(\boldsymbol{x}_i)[1] \right] \\
&+ 4 \sqrt{\frac{D_{\text{KL}}(q_\theta \| P_1) + D_{\text{KL}}(q_\omega \| P_2) + \log(\frac{6m}{\delta})}{m - 1}}
\end{aligned}
\tag{31}
$$

By the definition of $g_\theta$: $g_\theta(\boldsymbol{x})[-1] = \max_{\boldsymbol{u} \in N_\epsilon(\boldsymbol{x})} v_\theta(\boldsymbol{u})$, $g_\theta(\boldsymbol{x})[1] = v_\theta(\boldsymbol{x})$. In addition, by the definition of $h_\omega$: $h_\omega(\boldsymbol{x})[-1] = 0$ and $h_\omega(\boldsymbol{x})[1] = f_\omega(\boldsymbol{x})$. Therefore, we can rephrase Eq. 31 as

follows:

$$\mathbb{E}_{\boldsymbol{x} \sim D} \mathbb{1} \left[ \max_{\boldsymbol{u} \in N_\epsilon(\boldsymbol{x})} v_\theta(\boldsymbol{u}) > v_\theta(\boldsymbol{x}) \text{ or } \text{sign}(f_\omega(\boldsymbol{x})) \neq 1 \right]$$

$$\leq \frac{1}{m} \sum_{i=1}^{m} \mathbb{1} \left[ \max_{\boldsymbol{u} \in N_\epsilon(\boldsymbol{x})} v_\theta(\boldsymbol{u}) - \gamma_1 > v_\theta(\boldsymbol{x}_i) \text{ or } \text{sign}(f_\omega(\boldsymbol{x}_i) - \gamma_2) \neq 1 \right] \qquad (32)$$

$$+ 4 \sqrt{\frac{D_{\text{KL}}(q_\theta || P_1) + D_{\text{KL}}(q_\omega || P_2) + \log(\frac{6m}{\delta})}{m-1}}$$

$\square$

*Proof of Lem. 2.* We apply Lem. 5 with priors $P_1, P_2$ and posteriors $q_\theta, q_\omega$, distributions similar the proof of Thm. 1 in (Neyshabur et al., 2018). In their proof, they show that for their selection of prior and posterior distributions: (1) $\mathbb{P}_{\theta' \sim q_\theta} \left[ \max_{\boldsymbol{x} \in \mathbb{X}} |v_\theta(\boldsymbol{x}) - v_\theta(\boldsymbol{x})| < \frac{\gamma_1}{4} \right] \geq \frac{1}{2}$ holds and (2) $D_{\text{KL}}(q_\theta || P) = \mathcal{O}(dr^2 B^2 q \log(rq) C(\theta)/\gamma_1^2)$. By taking $\gamma_1$ to be half of the value the use and therefore, $\sigma$ (from their proof) to be half of the value they use as well, we obtain $\mathbb{P}_{\theta' \sim q_\theta} \left[ \max_{\boldsymbol{x} \in \mathbb{X}} |v_\theta(\boldsymbol{x}) - v_\theta(\boldsymbol{x})| < \frac{\gamma_1/2}{4} \right] \geq \frac{1}{\sqrt{2}}$ and $D_{\text{KL}}(q_\theta || P) = \mathcal{O}(dr^2 B^2 q \log(rq) C(\theta)/\gamma_1^2)$. We select $P_2$ and $q_\omega$ in a similar fashion. In particular, we can replace the penalty term in Lem. 5 as follows:

$$4 \sqrt{\frac{D_{\text{KL}}(q_\theta || P_1) + D_{\text{KL}}(q_\omega || P_2) + \log\left(\frac{6m}{\delta}\right)}{m-1}}$$

$$\in \mathcal{O} \left( \sqrt{\frac{B^2(r^2 q_1 \log(rq_1) C(v_\theta)/\gamma_1^2 + s^2 q_2 \log(sq_2) C(f_\omega)/\gamma_2^2) + \log\left(\frac{m}{\delta}\right)}{m}} \right) \qquad (33)$$

$\square$

## C ADDITIONAL FIGURES

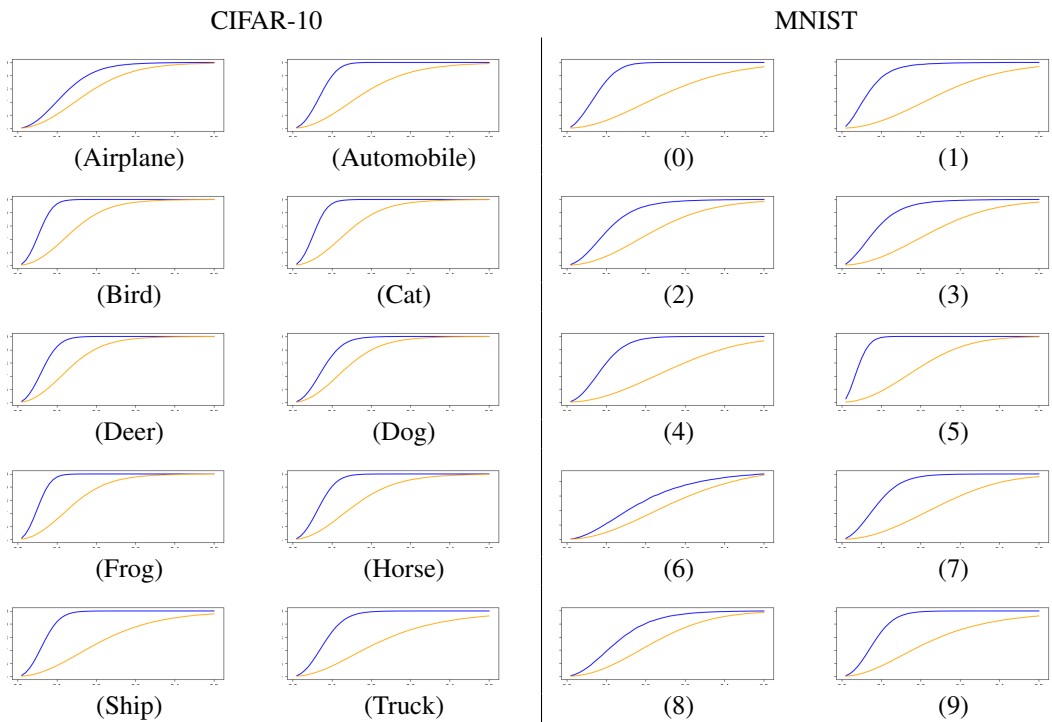

Figure 3: Same as Fig. 1, but where the images are taken from the test set of all classes, regardless of the single class used for training.

