# OpenReview forum: "Unsupervised Learning of the Set of Local Maxima"
_ICLR.cc/2019/Conference_

### Official Review · AnonReviewer3 · 2018-10-31
**hard to follow**

**Rating:** 8
**Confidence:** 3

**Review:**

The reviewer feels that the paper is hard to follow. The abstract is confusing enough and raises a number of questions.  The paper talks about `"local maxima" without defining an optimization problem. What is the optimization problem are we talking about? Is it a maximization problem or minimization problem? If we are dealing with a minimization problem, why do we care about maxima?

The first several paragraphs did not make the problem of interest clearer. But at least the fourth paragraph starts talking about training networks (the reviewer guesses this "network" refers to neural network, not other types network (e.g., Bayesian network) arising in machine learning). This paragraph talks about random initialization for minimizing a loss function, does this mean we are considering a minimization problem's local maxima? In addition, random initialization-based neural network training algorithms like back propagation cannot guarantee giving local maxima or local minima of the problem of interest (which is the loss function for training). It is even not clear if a stationary point can be achieved. So if the method in this paper wishes to work with local maxima of an optimization problem, this may not be a proper example.

The next paragraph brings out a notion of value function, which is hard to follow what it is. A suggestion is to give a much more concrete example to enlighten the readers.

The next two paragraphs seem to be very disconnected. It is not properly defined what is x and how to obtain it. If they are local maxima of a problem, please give us an example: what is the optimization problem, and why this is an interesting setup?

Since the problem setup of this paper is very hard to decode, it is also very hard to appreciate why the papers in the "related work" section are really related.

The motivation and intuition behind the formulations in (1) and (2) are hard to follow, perhaps because the goal and objective of the paper is unclear.

Overall, there is no formal problem definition or statement, and the notions and terminologies in this paper are not properly defined or introduced. This makes evaluating this work very hard.


========= after author feedback =======
After discussing with the authors through OpenReview, the reviewer feels that a lot of things have been clarified. The paper is interesting in its setting, and seems to be useful in different applications.  The clarity can still be improved, but this might be more of a style matter.  The analysis part is a bit heavy and overwhelming and not very insightful at this moment. Overall, the reviewer appreciate the effort for improving the readability of the paper and would like to change the recommendation to ````   accept.

---

> ### Author Response · Authors · 2018-11-06
> **definitions**
>
> It is always the authors’ responsibility that the readers understand their work. To maximize the probability that our work would be well understood, we have collected feedback from quite a few readers. Yet it seems that there is still room for improvement.
>
> While taking responsibility for this, we respectfully disagree with the claim of the reviewer that “there is no formal problem definition or statement, and the notions and terminologies in this paper are not properly defined or introduced.“  As can be seen, we define the problem we study multiple times: (i) it is defined clearly in the abstract (input, goal, which functions are learned, why, and how). (ii) it is defined again at the end of the introduction in the first three paragraphs of page 2. (iii) it is redefined again at the beginning of Sec. 3, since we were worried that some readers would skip the abstract and the introduction.
>
> Paragraphs 1-4 motivate our methods, by showing sample sets that arise in biology, man-made constructs, and weights of neural networks. The underlying value function in each case is explained: fitness or energetic efficiency in biology, an implicit value function in architecture (we mention a few possible factors), and an engineered loss in machine learning.
>
> It seems that the reviewer was confused by the last example since it discusses machine learning. However, the paragraph merely describes a process that generates unsupervised samples that are the result of a local optimization process.  The implication is that similarly to the other examples, viewing the learned weights of each random initialization as points in a vector space, this set of vectors is a suitable input to our method.
>
> It is emphasized in the abstract and then in the introduction that the value function is learned and that the local maxima are of that unlearned function. The paper starts with “[the] input is a set of unlabeled points that are assumed to be *local maxima of an unknown value function* in an unknown subset of the vector space”.
>
> The reviewer states that we discuss local maxima without stating the optimization problem. However, the local maxima we consider are of a function we seek to learn, not of an optimization problem. The notion of local maxima is discussed in the abstract, as it is actually applied: we learn a function h that compares the value of two points and a local maxima is a point x such that every point x’ in the vicinity of x satisfies c(x’) = -1 or is deemed by h to be lower in value than x.
>
> The notion of local maxima is also clearly defined in the intro: “In addition, we also consider a value function v, and for every point x', such that ||x' −x|| < eps, for a sufficiently small  eps > 0, we have: v(x0) < v(x)”. In practice, as mentioned early on in Sec. 3, and is well motivated by the ambiguity of v, we learn a comparator function h and not v.
>
> The reviewer says that “It is not properly defined what is x and how to obtain it”. The points x are the training samples and the definition of x is also given multiple times:
> (1)  The abstract says “all training samples x”.
> (2)  The introduction says that the points x are in the set S, which is defined as “Let S be the set of such samples from a space X”. The word “such” clearly refers in this context to unlabeled training samples.
> (3) This is repeated one paragraph below, at the beginning of Sec. 2, “The input to our method is a set of unlabeled points.”
> (4) As mentioned, we redefine x and the other concepts as soon as Sec. 3 starts, to make sure that all readers are aware of the setting. “Recall that S is the set of unlabeled training samples, and that we seek two functions c and v such that for all x \in S it holds that: (i) c(x) = 1, and (ii) x is a local maxima of v.” By “seek” we mean learn, but since it is not the first time this is stated in the paper (even the previous paragraph mentions that the value function is learned), we used a different word.
>
> The reviewer says that “The motivation and intuition behind the formulations in (1) and (2) are hard to follow, perhaps because the goal and objective of the paper is unclear. “. However, the terms of both equations are discussed one by one below them. These explanations are directly tied to the goals and objectives that appear earlier in the paper:
> (1) In the abstract: “Loss terms are used to ensure that all training samples x are a local maxima, according to h and satisfy c(x) = 1. Therefore, c and h provide training signals to each other: a point x’ in the vicinity of x satisfies c(x’) = −1 or is deemed by h to be lower in value than x. “
> (2) In the intro: “This structure leads to a co-training of v and c, such that every point x’ in the vicinity of x can be used either to apply the constraint v(x’) < v(x) on v, or as a negative training sample for c. Which constraint to apply, depends on the other function: if c(x’) = 1, then the first constraint applies; if v(x’) >= v(x), then x’ is a negative sample for c”.

---

### Official Review · AnonReviewer2 · 2018-11-01
**Interesting idea of casting one-class classification/set beloning problem onto 4 player game**

**Rating:** 8
**Confidence:** 4

**Review:**

This paper describes a new form of one-class/set beloning learning, based on definition of 4 player game:
- Classifier player (c), which is a typical one-class classifier model
- Comparator player (h), which given two instances answers if first is "not smaller" (wrt. set belonging) than the other
- Classifier adversary player (Gc), which tries to produce hard to distinguish samples for (c)
- Comparator adversary player (Gh), which tries to produce hard to classify samples for (h)
This way authors end up with cooperative-competitive game, where c and h act cooperatively to solve the problem, while Gc and Gh constantly try to "beat" them.

Overall I find this paper to be interesting and worth presenting, however I strongly encourage authors to rethink the way story is presented so that it is more approachable by people who do not have much experience with viewing typical classification problems as games. In particular, one could completely avoid talking about "sets of local maxima" and just talk about the density estimation problem, with c being characteristic function (of belonging to the support) and h being comparator of the pdf.

Strong points:
- Novel, multi-agent in nature, approach to one-class classification
- Proposed method build a complex system, which can be used in much wider class of problems than just classification (due to joint optimisation of classifier and comparator)
- Extensive evaluation on 4 problems
- Nice ablation study showing that most of the benefits come from pure c/Gc game (on average 68.8% acc vs 65.2% of just c, and 69.8% of entire system) but that h/Gh players do indeed still improve (an extra 1%). It might be interesting to investigate what exactly changed in c due to existance of h in training. Are there any identifiable properties of the model that can now be analysed?

Weak points:
In general I believe that theoretical analysis is the weakest part of the paper, and while interesting - it is actually a minor point, and shows interesting properties, but not the ones that would guarantee anything in "practical setup". I would suggest "downplaying" this part of the paper, maybe moving most of it to the appendix.
To be more specific:
- Theorem 1 shows that representation can be more compact, however existance of compactness does not rely imply that this particular solution can ever be learned or that it is a good thing (number of parameters is not correlated with generalisation capabilities of the model).
- Lemma 1 seems a bit redundant for the story. While it is nice to be able to show generalisation bounds in general, this paper is not really introducing new class of models (since in the end c is going to be used for actual classification), but rather training regime, and generalisations bounds do not tell us anything about the emerging dynamical system. The fact that adding v does not constrain c too much seems quite obvious, and as a result I would suggest moving this section to appendix.
Instead, if possible, the actual tricky mathematical bit for methods like this would be, in reviewers opinion, any analysis of learning dynamics of the system like this. Multi-agent systems cannot be optimised with independent gradient descent in general (convergence guarantees are lost). Consequently many papers focus on methods that bring these properties back (e.g. Consensus Optimisation or Symplectic Gradient Ascent). It would be beneficial for the reader to spend some time discussing stability of the system proposed, even if only empirically and on small problems.

Other remarks:
- eq. (1) is missing \cdot
- it could be useful to include explicit parameters dependences in (1) and (2) so that one sees how losses really define asymmetric game between the players
- why do we need 4 players and not just 3, with Gc and Gh being a single player/neural network? can we consider this as another ablation?
- given small performance gaps in Table 1 can we get error estimates/confidence intervals there? Deep SVDD paper includes error estimates of the baseline methods
- since training is performed in mini batch (it does not have to be decomposible over samples) shouldn't equations be based on expectations rather than sums?

-

---

> ### Author Response · Authors · 2018-11-14
> **Thank you for your support and the insightful comments**
>
> Thank you very much for the supportive and very detailed review.
>
> You suggest to reposition the paper as a density estimation problem. After much consideration we decided that a more conservative approach, in which we leave the current presentation and add the new viewpoint, would serve us better at this point. Your exciting perspective is now added to the introduction and we already received a positive feedback on it from AnonReviewer3.
>
> Following your suggestion, we have moved the theoretical part to the appendix. One small remark -- going forward, and applying the dual model beyond unsupervised learning, we expect h to become more dominant than c. For example, we are exploring an event detection model where the events occur at the local maxima of h, in regions that are defined by c.
>
> Reviewer: Multi-agent systems cannot be optimized with independent gradient descent in general (convergence guarantees are lost). Consequently many papers focus on methods that bring these properties back (e.g. Consensus Optimization or Symplectic Gradient Ascent). It would be beneficial for the reader to spend some time discussing stability of the system proposed, even if only empirically and on small problems.
>
>
> Answer: Following the review, we became familiar with the field of convergence of multi-agent systems. Thank you for pointing us in this direction. Our method could benefit in the future from the increased stability and theoretical guarantees one can obtain with these emerging methods.
>
> As requested, we tried to evaluate this empirically. We took the example from [1] of a mixture of 16 Gaussians that are placed on a 4x4 grid and applied our method, as well as variations in which we trained only c or only h. Since our method is meant to model local maxima and not entire high-probability regions, we take a standard deviation that is ten times smaller than previous work. These results, which can be found in the latest revised version, indicate that when jointly training c and h, the former captures all the 16 modes, and h is also informative. When training each alone, training results with mode hopping.
>
> [1] D. Balduzzi, S. Racaniere, J. Martens, J. Foerster, K. Tuyls, and T. Graepel. The Mechanics of n-Player Differentiable Games. ICML, 2018.
>
> To the other comments:
>
> 1. The \cdot was added to Eq. 1
>
> 2. Trying to add the parameters dependencies in Eq. 1 and 2 resulted in a cumbersome formulation. We therefore choose address the dependencies with added text.  Please let us know if you still prefer that we separate the equations.
>
> 3. A three player game is explored in two ways in the ablation of Tab.2: (i) In the lines that say “with G_c only” we use only G_c to generate negative points to both c and h and report results for both of these functions, and (ii) Same for “with G_h only”, where G_h was used to generate negative points to both networks. We altered the text to better reflect this.
>
> 4. We have added standard deviations to Tab. 1, similarly to the paper from which the baselines were taken. The results reported were already averaged over multiple runs.
>
> 5. Expectations rather than sums -- Following the suggestion, we have replaced the sum with averages. Writing the equations as expectations would require the addition of slightly more terminology and we wish to avoid this. Note that while SGD is indeed used, every step of Alg. 1 is over the entire samples of the training set (since the training sets are small).

---

> > ### Comment · AnonReviewer2 · 2018-11-18
> > **Thanks for revision**
> >
> > Thank you for updating the paper and providing missing information. Wrt. point 2, I am  fine with current formulation. I find the empirical results of lack of mode hopping intriguing, and would strongly suggest taking a deeper look into this phenomenon in the future. For the time being, I am increasing the score to 8, as the paper presentation (and results) significantly improved, and I believe it is a really solid work.

---

### Official Review · AnonReviewer1 · 2018-11-06
**Unsupervised Learning of the Set of Local Maxima**

**Rating:** 8
**Confidence:** 3

**Review:**

In this paper, the authors focus on the task of learning the value function and the constraints in unsupervised case. Different from the conventional classification-based approach, the proposed algorithm uses local maxima as an indicator function. The functions c and h and two corresponding generators are trained in an adversarial way. Besides, the authors analyzed that the proposed algorithm is more efficient than the conventional classification-based approach, and a suitable generalization bound is given. Overall, this work is theoretically complete and experimentally sufficient.
1.	The trained c and h give different predictions in most cases. As a unsupervised method, how to deal with them?
2.	In Table3, why can h achieve better results when adding noise?

---

> ### Author Response · Authors · 2018-11-08
> **Thank you for your comments**
>
> Thank you very much for your comments.
>
> It is true that c and h are trained concurrently and that the training algorithm, presented as Algorithm 1, is almost symmetric between the two. However, the two networks differ for multiple reasons: (i) The structure of the two functions is different: c has one input, and h has two, and (ii) The loss is different: G_h, which is the network that generates negative points for h, generates points G_h(x) that are in the vicinity of point x.
>
> These two differences are enough to ensure that h and c take different roles: c is, what AnonReviewer2 calls a characteristic function (does x belong to the set), and h is a comparator of nearby points.
>
> When there are multiple aspects that define the given set of input points, e.g., class membership and quality, c and h would assume the role that fits their structure, and not a random role.
>
> In addition, due to their loss, h and c strive to become anti-correlated, which further pushes them to take different roles. As mentioned, these roles are not arbitrary but depend on the structure of the two functions
>
> In the revision we uploaded earlier today, we put an additional emphasis on this asymmetry.
>
> To your questions #1:
>
> We use either c or h based on our goal. If, for the image experiments, our goal is to detect out-of-class samples, we use c. If our goal is to detect low quality images, we use h. In the cancer dataset experiment, h is more suitable for predicting the continuous value of survival we are interested with. A hypothetical scenario in which h and c play a different role in drug discovery is mentioned, for illustration, at the end of the discussion section.
>
> To your questions #2:
>
> The results in Tab. 3 are reported for multiple experiments, which are given side by side for brevity. In the columns of the experiment “(i) class membership” we evaluate the typical one-class classification scenario, for which c is suitable.
>
> In the other two scenarios, we test images from the training class vs. noisy images. In the experiment “(ii) Noise in-class” we evaluate the ability of each learned method to discriminate between images that are similar to those in the training set and images that are noisy versions of it. In this task, which is based on image quality, h, as a comparator, is more suitable.
>
> To see why this is the case, consider the training of h, during which points x are compared with generated points x’ in the vicinity of x. Since the training points x are obtained from a set of real-world training images, they are likely to be of higher quality than the generated nearby points.

---

### Public Comment · (anonymous) · 2018-12-01
**There are already superior results than yours for one-class classification**

Your results for one-class classification in Table 1 for CIFAR-10 are significantly inferior to the state-of-the-art. See the following NIPS paper:
http://papers.nips.cc/paper/8183-deep-anomaly-detection-using-geometric-transformations.pdf
Table 1 (page 9) in that paper shows outstandingly better results. Moreover, the performance of their algorithm is better for each and every class. Your average AUC for all CIFAR-10 experiments is 69.8, and the best known average AUC in that paper is 86.0.
Given these results, and the very high ratings of your paper, it is crucial to include the best known numbers (in the NIPS paper) in your Table 1.
We doubt it that the reviewers had given your paper such high ratings had they known about the state-of-the-art.

---

> ### Comment · AnonReviewer2 · 2018-12-02
> **Sticking to original rating**
>
> Thank you for pointing out this really interesting work.
>
> I am aware of this paper, and don't view it as in any sense - reducing quality of the paper under review, and as a reviewer - I am sticking to the currently assigned rating (8).
>
> While it might be interesting to point readers to the NIPS work in this paper, they are completely incomparable contributions. NIPS work is an image specific method, which focuses on data augmentation (to be more precise: enforced predefined geometrical transformation invariance), while paper under review is a generic scheme which happens to be applicable to one-class classification. Both methods seem orthogonal, and would be great to see them combined in some future work.

---

> ### Author Response · Authors · 2018-12-06
> **Golan and El-Yaniv, NIPS 2018**
>
> Thank you very much for pointing us to the NIPS 2018 work by Golan and El-Yaniv, which we will happily include in our next version.
>
> We completely agree with AnonReviewer2 that the two methods are different in their scope and orthogonal in their contributions and are working on combining both methods. It will take more than a few days, since the implementations of the two methods were written in different frameworks.

---

### Meta-Review · Area_Chair1 · 2018-12-12
**Novel work**

**Confidence:** 4
**Recommendation:** Accept (Poster)

**Metareview:**

The paper proposes a new unsupervised learning scheme via utilizing local maxima as an indicator function.

The reviewers and AC note the novelty of this paper and good empirical justifications. Hence, AC decided to recommend acceptance.

However, AC thinks the readability of the paper can be improved.